# Structure-guided microbial targeting of antistaphylococcal prodrugs

Justin J Miller[1,2], Ishaan T Shah[1], Jayda Hatten[1], Yasaman Barekatain[3], Elizabeth A Mueller[2], Ahmed M Moustafa[4], Rachel L Edwards[1], Cynthia S Dowd[5], Geoffrey C Hoops[6], R Jeremy Johnson[6], Paul J Planet[4], Florian L Muller[3], Joseph M Jez[2], Audrey R Odom John[1,4,7]*

[1]Department of Pediatrics, Washington University School of Medicine, St. Louis, United States; [2]Department of Biology, Washington University in St. Louis, St. Louis, United States; [3]Department of Cancer Systems Imaging, The University of Texas MD Anderson Cancer Center, Houston, United States; [4]Department of Pediatrics, Perelman School of Medicine, University of Pennsylvania, Children's Hospital of Philadelphia, Philadelphia, United States; [5]Department of Chemistry, The George Washington University, Washington, United States; [6]Department of Chemistry and Biochemistry, Butler University, Indianapolis, United States; [7]Department of Molecular Microbiology, Washington University School of Medicine, St. Louis, United States

**Abstract** Carboxy ester prodrugs are widely employed to increase oral absorption and potency of phosphonate antibiotics. Prodrugging can mask problematic chemical features that prevent cellular uptake and may enable tissue-specific compound delivery. However, many carboxy ester promoieties are rapidly hydrolyzed by serum esterases, limiting their therapeutic potential. While carboxy ester-based prodrug targeting is feasible, it has seen limited use in microbes as microbial esterase-specific promoieties have not been described. Here we identify the bacterial esterases, GloB and FrmB, that activate carboxy ester prodrugs in *Staphylococcus aureus*. Additionally, we determine the substrate specificities for FrmB and GloB and demonstrate the structural basis of these preferences. Finally, we establish the carboxy ester substrate specificities of human and mouse sera, ultimately identifying several promoieties likely to be serum esterase-resistant and microbially labile. These studies will enable structure-guided design of antistaphylococcal promoieties and expand the range of molecules to target staphylococcal pathogens.

*For correspondence:
johna3@email.chop.edu

Competing interests: The authors declare that no competing interests exist.

## Introduction

Antimicrobial resistance presents a major challenge to public health (*Hsu, 2020*; *Antimicrobial resistance in the age of COVID-19, 2020*). In 2019, 2.8 million antibiotic-resistant infections occurred in the United States, resulting in 35,000 deaths (*CDC, 2019*). Some estimates have suggested that antimicrobial-resistant infections will cause as many as 10 million deaths annually by 2050 (*WHO (World Health Organisation), 2019a*). *Staphylococcus aureus* is a devastating human pathogen. Methicillin-resistant *S. aureus* (MRSA) has been labeled a 'serious threat' by the Centers for Disease Control and Prevention (*CDC, 2019*; *Kourtis et al., 2019*; *Turner et al., 2019*). New antimicrobials, especially those with novel mechanisms of action, are urgently needed; however, most anti-infectives under development take advantage of existing antibiotic scaffolds with proven efficacy and established safety profiles (*Global Antimicrobial Resistance Hub, 2020*; *WHO (World Health Organisation), 2019b*). New, chemically distinct antibiotics are highly desirable as a strategy to circumvent antimicrobial resistance.

Many metabolic processes are essential for microbial growth and/or pathogenesis, although few existing antimicrobials exploit metabolism as an anti-infective target. Metabolic drug design can be facile, as natural substrates serve as a template for competitive inhibitors. As metabolism often involves the transformation of highly charged and polar metabolites, inhibitors of metabolic enzymes often take advantage of phosphonate functional groups for target engagement (*Azema et al., 2006*). Unfortunately, negatively charged phosphonates are readily excluded from cell membranes and often exhibit poor drug-like properties (*Edwards et al., 2020*; *Hecker and Erion, 2008*; *Heidel and Dowd, 2019*; *Hsiao et al., 2014*; *Kornberg et al., 1972*; *Lin et al., 2020*; *Mackie et al., 2012*; *McKenney et al., 2012*; *Wiemer and Wiemer, 2015*; *Zhang et al., 2006*). New strategies enabling effective deployment of antimetabolites will serve to expand the druggable space for antimicrobials.

One means of improving phosphonate cellular permeability is to chemically mask the undesirable negative charge with lipophilic groups. This strategy (termed prodrugging) can employ labile pro-moieties that are removed during absorption, distribution, or intracellularly to yield the original phosphonate antibiotic (*Figure 1A*; *Hecker and Erion, 2008*; *Heidel and Dowd, 2019*; *Wiemer and Wiemer, 2015*). We have previously designed a series of MEPicide antibiotics, in which phosphonate isoprenoid biosynthesis inhibitors are modified with a lipophilic pivaloyloxymethyl (POM) promoiety. MEPicide prodrugs bypass the need for active cellular transport, while simultaneously increasing compound potency against zoonotic staphylococci, *S. schleiferi* and *S. pseudin-termedius*, as well as *Plasmodium falciparum* and *Mycobacterium tuberculosis* (*Edwards et al., 2020*; *Edwards et al., 2017*; *McKenney et al., 2012*; *San Jose et al., 2016*; *San Jose et al., 2013*; *Uh et al., 2011*). However, POM promoieties are rapidly hydrolyzed by serum carboxylesterases, which limits the in vivo efficacy of POM prodrugs as a strategy to improve the drug-like features of charged antimicrobials (*Lin et al., 2020*; *Wang et al., 2018*).

To enable effective cellular delivery of phosphonate antibiotics, new lipophilic prodrug moieties that are resistant to serum carboxylesterases – yet cleavable by microbial esterases – are required.

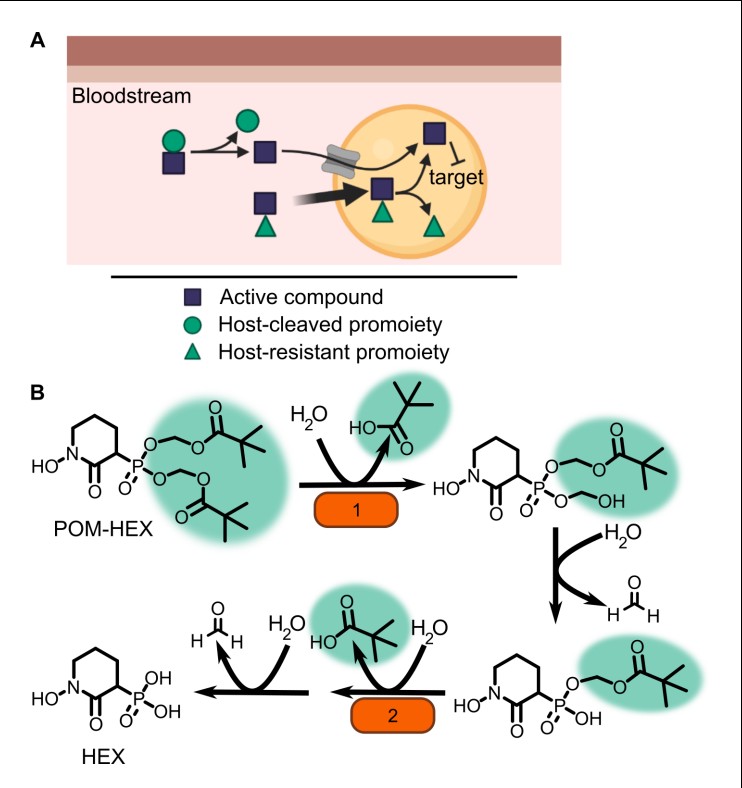

**Figure 1.** Prodrug activation model (A) and proposed enzymatic mechanism. (B) Carboxy ester promoieties highlighted in green.

This strategy has been effectively used to achieve tissue-specific selective prodrug activation. For example, liver-targeted prodrug delivery was achieved by harnessing the distinct substrate specificity of the liver-specific isoform of P450, CYP3A4 (*Erion et al., 2005*; *Erion et al., 2004*). Accordingly, understanding the enzymatic and structural mechanisms of microbial prodrug activation will facilitate the development of microbe-specific prodrugs.

We recently described the staphylococcal enzyme, GloB, which facilitates activation of carboxy ester prodrugs in *S. schleiferi* and *S. pseudintermedius* (*Mikati et al., 2020*). Notably, GloB is insufficient to fully activate prodrugs in vitro, suggesting that at least one additional enzyme is necessary for complete phosphonate release. In this work, we identify and characterize the role of two *S. aureus* esterases, GloB and FrmB, which each catalyze carboxy ester prodrug hydrolysis and contribute to carboxy ester prodrug activation. We demonstrate that both esterases have defined substrate specificities, which diverge from the substrate specificities of human and mouse sera. Finally, we present the three-dimensional structures of GloB and FrmB to enable ongoing structure-guided design of FrmB- and GloB-targeted prodrugs.

## Results

### Identification of microbial esterases responsible for carboxylesterase activity

In the zoonotic staphylococcal species *S. schleiferi* and *S. pseudintermedius*, loss of the enzyme GloB, a hydroxyacylglutathione hydrolase, or glyoxalase II enzyme confers resistance to carboxy ester prodrugs because carboxy ester prodrugs are not activated (*Mikati et al., 2020*). However, purified GloB alone is insufficient to activate carboxy ester prodrugs in vitro, suggesting that at least one additional cellular enzyme is required. Based on the hypothesized carboxy ester activation pathway, we predicted that the missing enzyme(s) might be either another carboxylesterase or a phosphodiesterase (*Figure 1B*). To establish the pathway for carboxy ester prodrug activation in *S. aureus*, we made use of the Nebraska Transposon Mutant Library (NTML), in which nearly 2000 nonessential *S. aureus* genes have been individually disrupted by a stable transposon insertion (*Fey et al., 2013*). Using the gene ontology feature on the NTML website (https://app1.unmc.edu/fgx/gene-ontologies.html), we identified 6 carboxylic ester hydrolases (including *gloB*), 11 phosphatases, and 9 phosphoric diester hydrolases as candidate activators of carboxy ester prodrugs (*Figure 2—source data 1*). Each identified transposon mutant was screened for resistance to the carboxy ester prodrug, POM-HEX. POM-HEX is a pivaloyloxymethyl prodrug of the compound HEX, which inhibits the glycolytic enzyme enolase (*Figures 1B* and *Figure 2A*). Of the 26 candidate esterase transposon mutants, only two strains were significantly more resistant to POM-HEX than the *S. aureus* parental strain, JE2, as determined by half-maximal growth inhibitory concentration (IC$_{50}$) (*Figure 2B*, *Figure 2—source data 1*). One of these strains had a transposon disrupting the gene that encodes the glyoxalase II enzyme, GloB. The GloB ortholog in *S. scheiferi* is a prodrug-activating enzyme, mutation in which confers resistance to POM-HEX (*Mikati et al., 2020*). The second POM-HEX-resistant strain harbored a transposon insertion in the locus encoding the predicted carboxylesterase annotated as FrmB. FrmB has been previously identified as FphF, a serine hydrolase, and is the primary *S. aureus* target of the sulfonyl fluoride compound JCP678 (*Lentz et al., 2018*). As *S*-formylglutathione hydrolase activity is more likely the biological function of this protein, we will refer to this protein as it has been named in *E. coli*, FrmB.

In parallel, we employed an unbiased forward genetics approach to identify genetic changes associated with POM-HEX resistance. POM-HEX-resistant staphylococci were derived by exposing wild-type (WT) *S. aureus* Newman to growth inhibitory concentrations (IC) of POM-HEX. In total, we selected and cloned 25 isolates with IC$_{50}$ values ranging from 1.5× to 16× that of WT *S. aureus* Newman (*Figure 2C,D*, *Figure 2—source data 2*).

Whole-genome sequencing of POM-HEX-resistant strains revealed mutations in *frmB* (n = 7), *gloB* (n = 10), and strains with no mutations in either *frmB* or *gloB* (n = 8). Most mutations were nonsynonymous single-nucleotide polymorphisms (SNPs) (*Figure 2D*, *Figure 2—source data 2*). In three instances, *gloB* was the only verified genetic change in the genome. Additionally, *frmB* and *gloB* each had one instance of a mutation resulting in a premature stop codon truncating the protein to less than a 100 amino acid sequence. Strains with no identified mutations in either *frmB* or *gloB* did

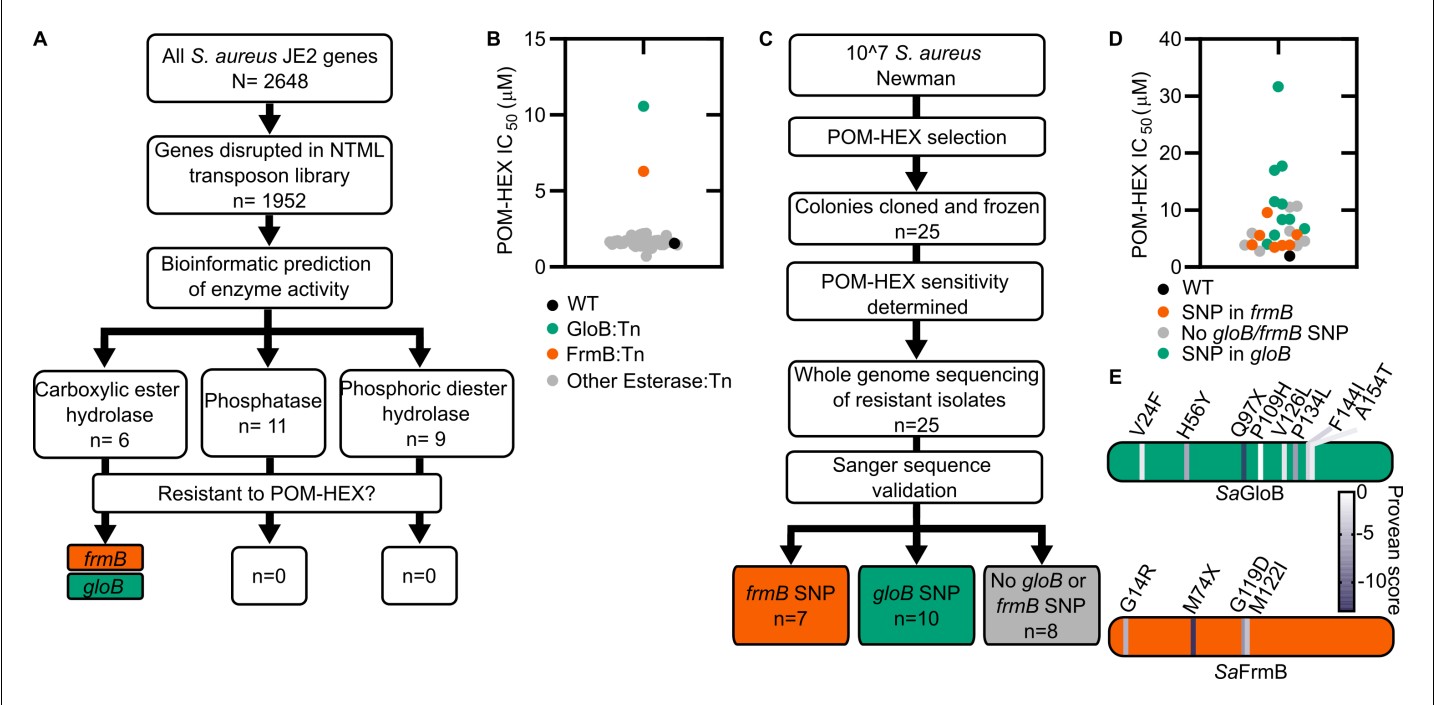

**Figure 2.** Forward and reverse genetics approaches identify FrmB and GloB as candidate POM-prodrug hydrolases in *S. aureus*. (A) Reverse genetics identification of candidate prodrug activating enzymes. (B) POM-HEX susceptibility of identified candidate resistance genes from (A) as determined by $IC_{50}$. Exact values and error reported in *Figure 2—source data 1*. (C) Forward genetic screen approach, all mutations listed in *Figure 2—source data 2*. (D) POM-HEX susceptibility of POM-HEX-resistant *S. aureus*. (E) Nonsynonymous point mutations identified by whole-genome sequencing in *frmB* and *gloB*. In all experiments, GloB is colored green and FrmB orange. Displayed are the means of three independent biological experiments. The online version of this article includes the following source data and figure supplement(s) for figure 2:

**Source data 1.** *S. aureus* transposon mutants tested and POM-HEX sensitivity.
**Source data 2.** *S. aureus* Newman resistant isolate SNPs and POM-HEX sensitivity.
**Source data 3.** *S. aureus* transposon mutants with genes identified by whole-genome sequencing and POM-HEX sensitivity.
**Source data 4.** Accession numbers for the isolates used in WhatsGNU analysis.
**Figure supplement 1.** Conservation of FrmB and GloB within *S. aureus*.
**Figure supplement 2.** Phylogenetic tree of FrmB and GloB.
**Figure supplement 3.** Enzymatic characterization of GloB and FrmB.
**Figure supplement 4.** NMR characterization of POM-HEX activation by GloB and FrmB.

not have any mutations, which are immediately obvious as POM-HEX resistance mechanisms. Of the non-*frmB* and non-*gloB* mutations, we selected transposon mutants from the NTML that had transposon insertions in the corresponding genes. These transposon mutations did not confer resistance to POM-HEX (*Figure 2—source data 3*). Overwhelmingly, the observed mutations in both *frmB* and *gloB* are predicted to have deleterious effects on protein function (PROVEAN score below a threshold of −2.5) (*Figure 2E*).

To evaluate the sequence conservation of FrmB and GloB among *S. aureus*, we performed a WhatsGNU analysis on all publicly available *S. aureus* genomes. WhatsGNU is a bioinformatic tool that can compress large databases and assess the number of instances in which a specific gene has 100% sequence coverage and identity match within the entire database (*Moustafa and Planet, 2020*). This parameter, the gene novelty unit (GNU) score, is high when a sequence is under strong selective pressure within the population and low when the gene is variable. GloB exhibits an exceptionally high GNU score of 8215 (of 10,350 possible), indicating that there is strong sequence conservation for *S. aureus* GloB. Conversely, FrmB sequences appear to be extremely conserved within individual *S. aureus* clonal complexes but varied between each complex (GNU scores of 2218 or 3370 of 10,350, *Figure 2—figure supplement 1*). We constructed a phylogenetic tree of GloB and FrmB orthologs among microbial populations. Most bacterial species contain a GloB ortholog,

though the primary sequence is highly variable across species and does not readily cluster according to the tree of life, as represented by the RNA polymerase RpoB (*Figure 2—figure supplement 2*). FrmB sequences are also highly sequence divergent, though they tend to cluster closer to the expected tree of life (*Figure 2—figure supplement 2*).

As new antibiotics are introduced, it is important to understand the potential for spontaneous resistance, as this can inform whether co-treatments may be required to circumvent resistance. We determined the resistance frequency for *S. aureus* Newman to POM-HEX as well as several clinically relevant antibiotics. Resistance frequencies were determined at 4 and 10 times the minimum inhibitory concentration (MIC) for each antibiotic in isolation. Spontaneous resistance readily developed during monotreatment with the DNA-dependent RNA synthesis inhibitor, rifampicin, which was used as a comparator (*O'Neill et al., 2001*). We find that spontaneous resistance to rifampicin occurs with frequencies of ~8E-8. Spontaneous resistance was not observed to the cell wall inhibitors nafcillin or vancomycin, nor in the protein synthesis inhibitors linezolid and clindamycin suggesting resistance frequencies of <1 E-11. Spontaneous resistance to POM-HEX occurs with frequencies between 2.9 and 6.0 E-7 (*Table 1*).

The agreement between our forward and reverse genetic screens strongly suggests that two discrete predicted esterases, rather than a pool of redundant cellular esterases, activates the POM-HEX prodrug in *S. aureus*. Additionally, the finding that mutation in either *frmB* or *gloB* is sufficient to confer POM-HEX resistance suggests that the two enzymes may work in concert to bioconvert POM-HEX into HEX. As substantial sequence variation occurs among *frmB* and *gloB*, any prodrug therapeutics, which hijack these two enzymes, may not be broadly activated across all bacteria.

## FrmB and GloB are carboxylesterases with diverging substrate specificity

GloB is predicted to be a type II glyoxalase and a member of the large metallo-β-lactamase protein superfamily (INTERPRO IPR001279). Glyoxalase II enzymes, including the closely related GloB ortholog from *S. scheiferi*, catalyze the second step in the glyoxalase pathway that is responsible for the cellular conversion of methylglyoxal (a toxic glycolytic byproduct) to lactic acid (*Melino et al., 1998*; *Mikati et al., 2020*; *Stamp et al., 2010*; *Zang et al., 2001*). Conversely, FrmB orthologs hydrolyze p-nitrophenyl esters of short-chain fatty acids (C2–C6) and are thought to mediate detoxification of cellular formaldehyde (*Fellner et al., 2020*; *Gonzalez et al., 2006*).

We purified recombinant WT *Sa*FrmB and *Sa*GloB to evaluate the enzymatic function of each protein (*Figure 2—figure supplement 3A*). We first assessed glyoxalase II activity using an assay, which couples hydrolysis of the glyoxalase II substrate, *S*-lactoylglutathione, to a change in absorbance (*Figure 2—figure supplement 3B*). *Sa*GloB hydrolyzes *S*-lactoylglutathione with a specific activity comparable to previously characterized microbial type II glyoxalases, but *Sa*FrmB lacks appreciable glyoxalase activity.

We next assessed the ability of FrmB and GloB to hydrolyze p-nitrophenyl esters of short-chain fatty acids that have a photometric change upon hydrolysis (*Figure 2—figure supplement 3C*). FrmB has modest activity against 4-nitrophenyl acetate and butyrate but no activity against 4-nitrophenyl trimethylacetate, suggesting a preference for unbranched fatty acids (*Figure 2—figure*

**Table 1.** Mutational frequencies of *S. aureus*.
Newman for POM-HEX and several clinically utilized antibiotics. Experiments performed in technical duplicate and biological triplicate. Displayed are the means ± SD.

| Antibiotic (MIC) | 4× MIC | 10× MIC |
|---|---|---|
| Vancomycin (2 µg/mL) | <1 E-10 | <1 E-10 |
| Clindamycin (0.25 µg/mL) | <1 E-10 | <1 E-10 |
| Rifampicin (7.5 ng/mL) | 8.96 ± 0.20 E-08 | 7.64 ± 2.10 E-08 |
| Nafcillin (0.5 µg/mL) | <1 E-10 | <1 E-10 |
| Linezolid (1.67 µg/mL) | <1 E-10 | <1 E-10 |
| POM-HEX (3.13 µg/mL) | 5.96 ± 1.60 E-07 | 2.92 ± 0.57 E-07 |

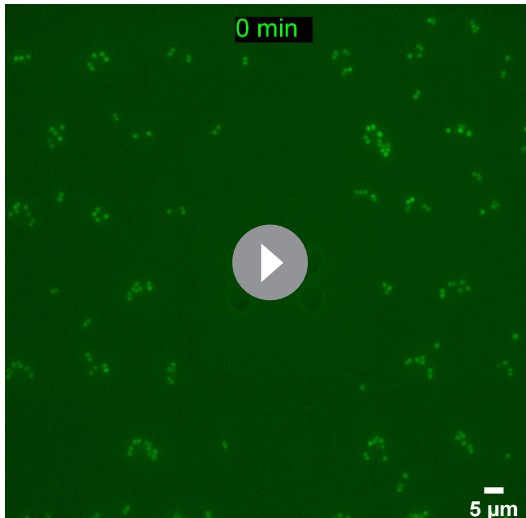

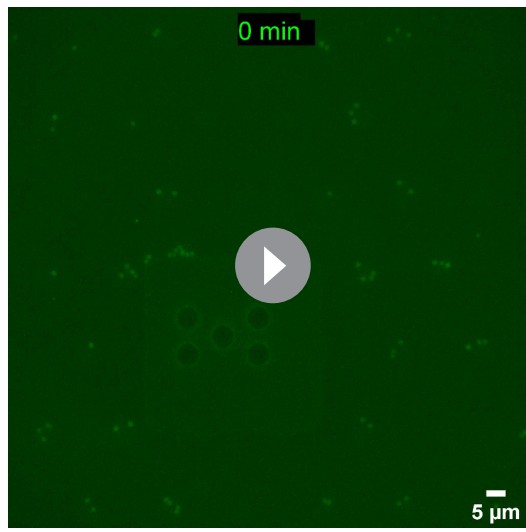

**Video 1.** In vivo activation rates depend on ester promoiety selection. Time series of fluorogenic ester substrate 1O activation. Substrate added at t = 10 min. Experiments were performed in biological duplicate.
https://elifesciences.org/articles/66657#video1

**Video 2.** In vivo activation rates depend on ester promoiety selection. Time series of fluorogenic ester substrate 3C activation. Substrate added at t = 10 min. Experiments were performed in biological duplicate.
https://elifesciences.org/articles/66657#video2

*supplement 3C*). This finding is in agreement with a previous characterization of FrmB as having a preference for short-chain and unbranched hydrophobic lipid substrates (*Fellner et al., 2020*). Notably, GloB has no detectable activity against these substrates and neither GloB nor FrmB hydrolyzes 4-nitrophenyl trimethylacetate despite its structural similarity to POM-HEX. This may be due to the absence of the acyloxymethyl ether moiety in 4-nitrophenyl substrates, which is found in POM-prodrugs.

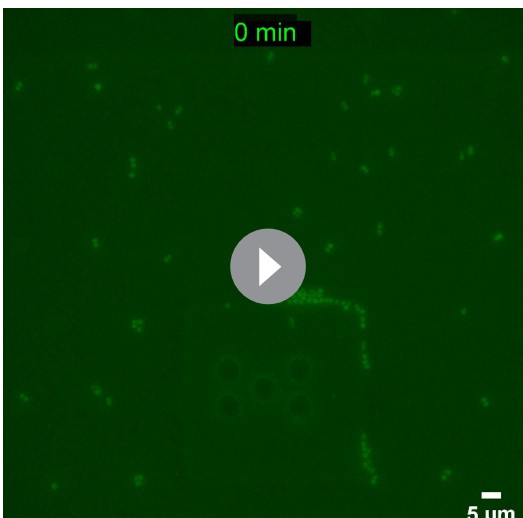

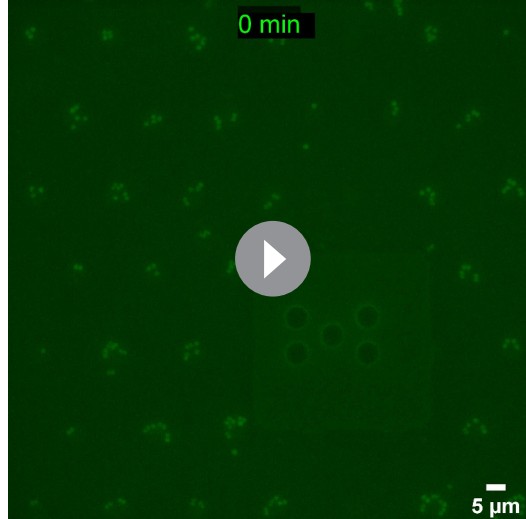

**Video 3.** In vivo activation rates depend on ester promoiety selection. Time series of fluorogenic ester substrate 5O activation. Substrate added at t = 10 min. Experiments were performed in biological duplicate.
https://elifesciences.org/articles/66657#video3

**Video 4.** In vivo activation rates depend on ester promoiety selection. Time series of fluorogenic ester substrate 9C activation. Substrate added at t = 10 min. Experiments were performed in biological duplicate.
https://elifesciences.org/articles/66657#video4

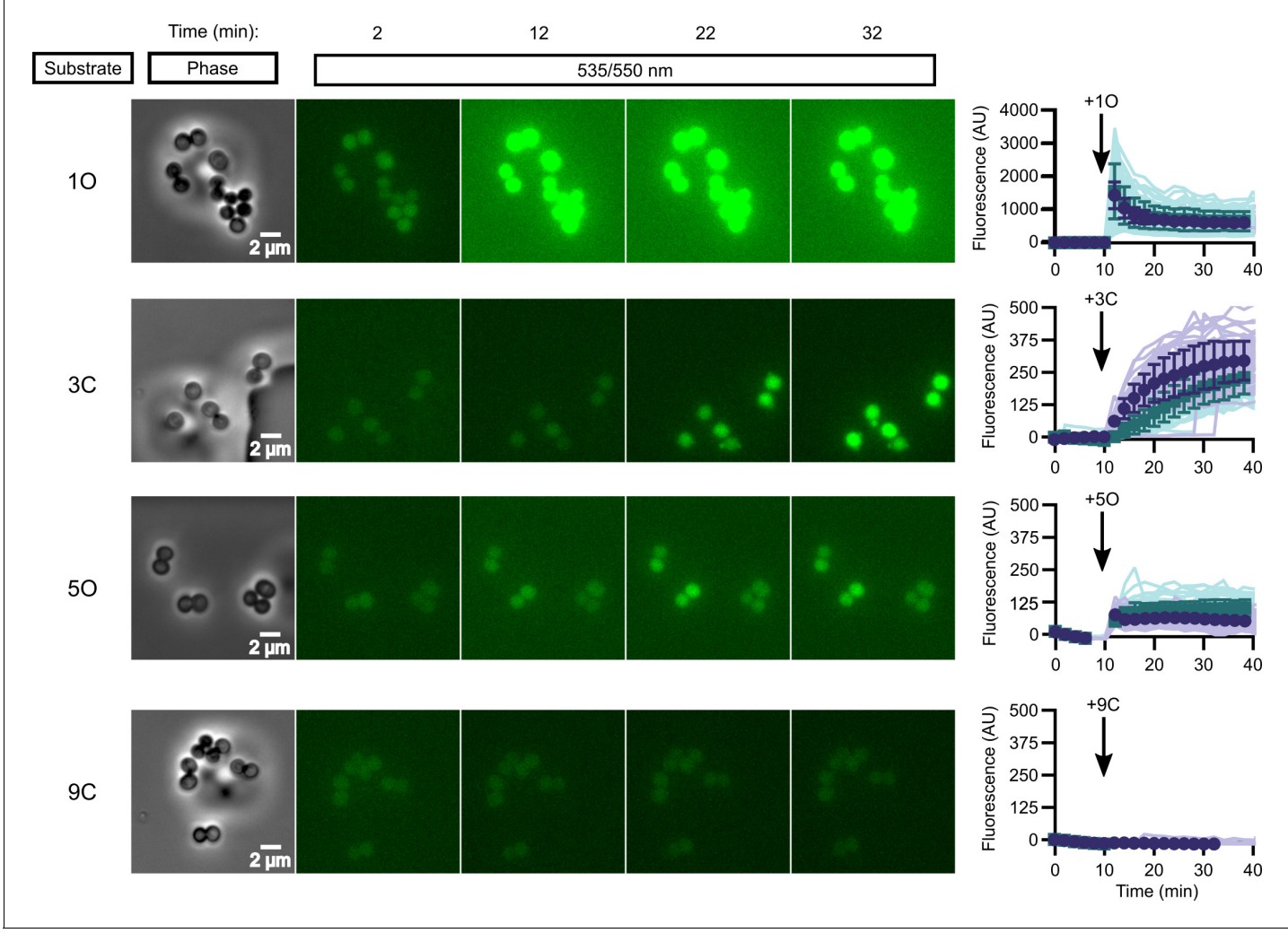

**Figure 3.** In vivo activation rates depend on ester promoiety selection. Time series of activation of various fluorogenic substrates (*Figure 3—figure supplement 1*). Substrates were added into the microfluidics chamber at t = 10 minutes. On the right, quantification of individual cell or cell cluster fluorescence per area. Faint traces are individual cells and darker traces represent the mean of a given experiment. Each experiment was performed in biological duplicate, and each experiment is displayed in a different color (purple or green). Full movies viewable as *Videos 1–4*. Error bars denote SD. The online version of this article includes the following source data and figure supplement(s) for figure 3:

**Source data 1.** (1) Michaelis–Menten parameters for *Sa*FrmB.
**Figure supplement 1.** Profluorescent substrate library.
**Figure supplement 2.** Catalytic efficiency of GloB (**A**) and FrmB (**B**).

We also sought to directly assess the role of GloB and FrmB in POM-HEX activation. We incubated each enzyme with POM-HEX and characterized the products via $^{31}$P-$^{1}$H-heteronuclear single quantum coherence (HSQC) nuclear magnetic resonance (NMR). We have previously shown that GloB removes only one POM moiety, resulting in an accumulation of mono-POM-HEX (*Figure 1B*; *Mikati et al., 2020*). Similarly, FrmB is capable of removing only one POM moiety (*Figure 2—figure supplement 4*). We hypothesized that the two esterases may be stereoselective and incubated both enzymes with POM-HEX. We find that incubation of POM-HEX with GloB and FrmB still results in an accumulation of mono-POM-HEX, suggesting that neither of the two esterases efficiently cleaves the charged mono-POM species (*Figure 2—figure supplement 4*).

## GloB and FrmB substrate specificity

To facilitate design of microbially targeted prodrug activation using these two enzymes, we next sought to extensively characterize GloB and FrmB substrate specificity. We employed a 32-compound ester substrate library, which fluoresces upon esterase activity (*Figure 3—figure supplement 1*; *White et al., 2018*). This library systematically varies ester substrate length, branching patterns, and ether and sulfide positioning, thereby allowing for the precise determination of structure-activity relationships. Kinetic measurements were performed for both FrmB and GloB over a range of substrate concentrations for the entire library, allowing for the calculation of catalytic efficiency ($k_{cat}/K_m$) for each enzyme and substrate (*Figure 3—source data 1* and 2).

We find that FrmB and GloB tend to have the highest activity toward oxygen ethers (*Figure 3—figure supplements 1* and *2*). GloB has the highest activity against short-chain ethers (compounds 1–3), with some tolerance for branching at the first carbon beyond the ester carbonyl (compounds 7–9), while extensive branching strongly reduces activity (compound 10). Remarkably, GloB is tolerant of the extreme steric bulk introduced with the phenoxyacetic acid substrate, if the substrate contains an oxygen or sulfur ether (compound series 11). GloB exhibits a strong preference for oxygen at the β-position to the carbonyl over the γ-position but is indifferent to the positioning of sulfur. While GloB has a wider range of catalytic specificities, FrmB exhibits lower overall activity and a narrower range of catalytic efficiency. FrmB hydrolyzes unbranched substrates with little regard for chain length or the end-of-chain bulk within the tested substrates (compound series 1–3, 11). Branching at the position following the ester carbonyl (compound series 7–9, 12) is deleterious to FrmB activity. When oxygen is included in the chain, positioning at the β-position to the carbonyl is strongly preferred over the γ-position. For both GloB and FrmB, expanding the library of substrates tested will be revelatory of the true limitations of enzymatic activity.

## Importance of substrate specificity in vivo

While in vitro enzymatic substrate profiling is informative for how individual enzymes activate prodrugs, it may not reflect the complex biochemical processes happening in vivo, where additional cellular esterases may impact overall compound activation. We performed quantitative live cell imaging to measure the activation of pro-fluorescent substrates in real time. We loaded *S. aureus* onto a microfluidic device and imaged the cells by phase-contrast and epifluorescence microscopy before and while supplying pro-fluorescent compound. Intracellular fluorescence accumulates in response to the rapid introduction of substrate into the chamber and can be quantified through time.

We selected four pro-fluorescent substrates of varying catalytic efficiency against FrmB and GloB to test in our microfluidics experiments. In vitro, substrate 1O displays high activity for both FrmB and GloB, 3C displays moderate activity for FrmB and GloB, 5O has moderate activity against GloB but poor activity against FrmB, and 9C has poor activity against both GloB and FrmB (*Figure 3—figure supplements 1* and *2*). Comparing the activation of these substrates through time, we find that our in vitro catalytic efficiency determination correlates well with our intracellular activation rates (*Figure 3*, *Videos 1–4*). Compound 1O, which exhibits high catalytic efficiency with both GloB and FrmB, reaches fluorescence saturation before the first 2 min time point. Compound 3C, which GloB and FrmB use with moderate activity, slowly activates over the duration of the experiment, and 5O and 9C, which have moderate-to-poor turnover with both GloB and FrmB, never appreciably activates during the 30 min of observation (*Figure 3*). As fluorescent activation is quantified per individual cell, we can also assess the uniformity of prodrug activation across the population. We observe remarkably homogenous activation of prodrugs across all observed cells (*Figure 3*).

## Three-dimensional structure of FrmB

To establish the structural basis for FrmB and GloB substrate specificity and enable future structure-guided prodrug therapeutic design, we solved the three-dimensional structures of both *S. aureus* FrmB and GloB. *S. aureus* FrmB was solved at 1.60 Å using molecular replacement with the low-temperature active alkaline esterase Est12 (PDB ID: 4RGY) as a search model (*Hu et al., 2015*). Refinement parameters and statistics are summarized in *Figure 4—source data 1*. A single dimer of FrmB is observable in the asymmetric unit, matching the apparent molecular weight of FrmB as observed via size-exclusion chromatography. The overall fold of FrmB is characteristic of the α/β-hydrolase fold. Six parallel β-strands and one anti-parallel β-strand pair form a central eight stranded β-sheet,

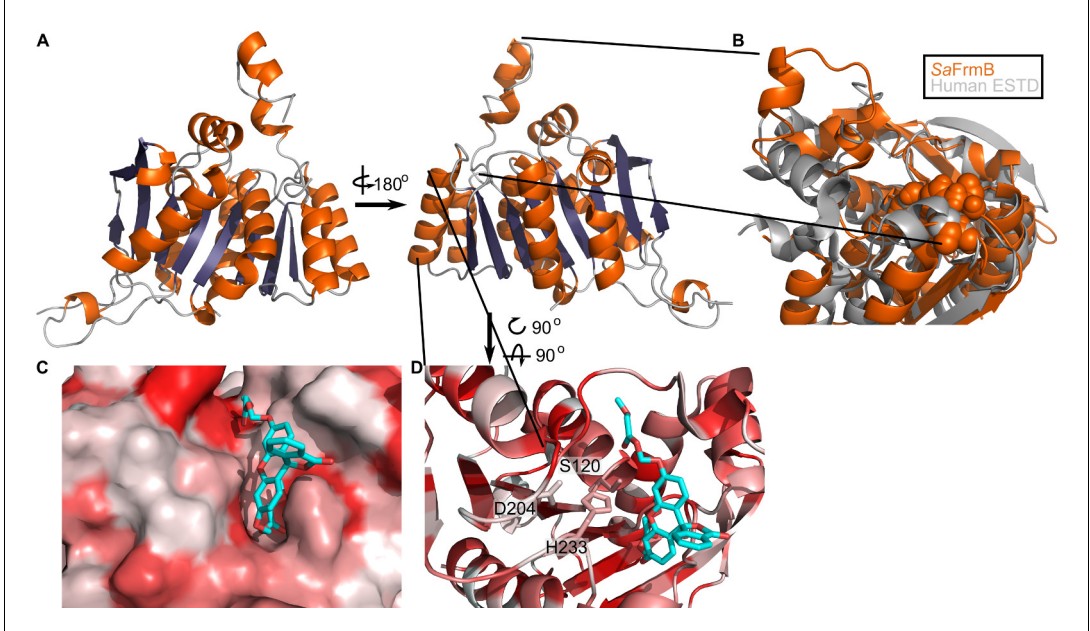

**Figure 4.** Three-dimensional structure of FrmB. (**A**) Overall fold, a-helices colored in orange and β-strands colored in purple. (**B**) Comparison between SaFrmB (orange) and its closest human ortholog, ESTD (gray). Active site residues denoted in orange spheres. (**C, D**) Docking of substrate 1O (sticks) in the active site of FrmB. surface view, red indicates highly hydrophobic and white hydrophilic residues. Surface view (**C**) or stick view with catalytic triad (**D**).

The online version of this article includes the following source data and figure supplement(s) for figure 4:

**Source data 1.** Summary of crystallographic data collection and refinement statistics.

**Figure supplement 1.** Structural conservation of FrmB.

which is surrounded by α-helices (*Figure 4A*). One monomer of FrmB has electron density for a single magnesium ion, whereas the second monomer has two magnesium ions present.

A structural similarity search was performed using the DALI server to identify proteins related to *Sa*FrmB. The structure of *Sa*FrmB was most similar to the molecular replacement model, Est12 from deep sea bacteria (PDB ID 4RGY, root mean squared deviation [r.m.s.d.] = 1.020 Å), but also had similarity to *Bacteroides intestinalis* ferulic acid esterase *Bi*Fae1A (PDB ID 5VOL, r.m.s.d. = 1.137 Å) and *Streptococcus pneumonia* tributyrin esterase estA (PDB ID 2UZ0, r.m.s.d. = 1.329 Å) (*Kim et al., 2007*; *Richardson et al., 2007*; *Wefers et al., 2017*). All structures display strong structural conservation including the positioning of the prototypic serine hydrolase catalytic triad: Ser120, Asp204, and His233 (*S. aureus*) (*Figure 4—figure supplement 1*). The most striking difference between the related structures is the flexible cap domain, implicated in substrate specificity of EstA and Est12 (*Hu et al., 2015*; *Kim et al., 2007*). While this manuscript was in preparation, an independent structure of FrmB was reported (*Fellner et al., 2020*). The two structures are nearly identical (PDB ID 6ZHD, r.m.s.d = 0.433 Å), with slightly different positioning of the capping domain.

We compared *Sa*FrmB to its closest human ortholog, human esterase D (PDB ID 3fcx), finding moderate structural similarity both in the overall fold (r.m.s.d = 4.625 Å) and in the positioning of the catalytic triad (*Wu et al., 2009*). However, *Sa*FrmB and human esterase D differ notably in the solvent-accessible surface around the active site, suggesting the potential for distinct substrate utilization, primarily driven by differential positioning of the cap domain (*Figure 4B*).

Using Autodock Vina, we modeled the highest catalytic efficiency substrate of FrmB, 1O, onto the active site of FrmB (*Morris et al., 2009*; *Trott and Olson, 2010*). Serine hydrolases classically bind the substrate carbonyl oxygen in an oxyanion hole, and substrate hydrolysis is initiated through attack of the catalytic serine on the ester carbonyl. The docking of 1O on FrmB mimics the initial state of a serine hydrolase reaction, with the carbonyl oxygen buried and the catalytic serine poised for attack (*Figure 4C*). The pocket directly next to the oxyanion hole is relatively narrow, suggesting that steric hindrance explains the poor activity of FrmB against branched substrates. The active site

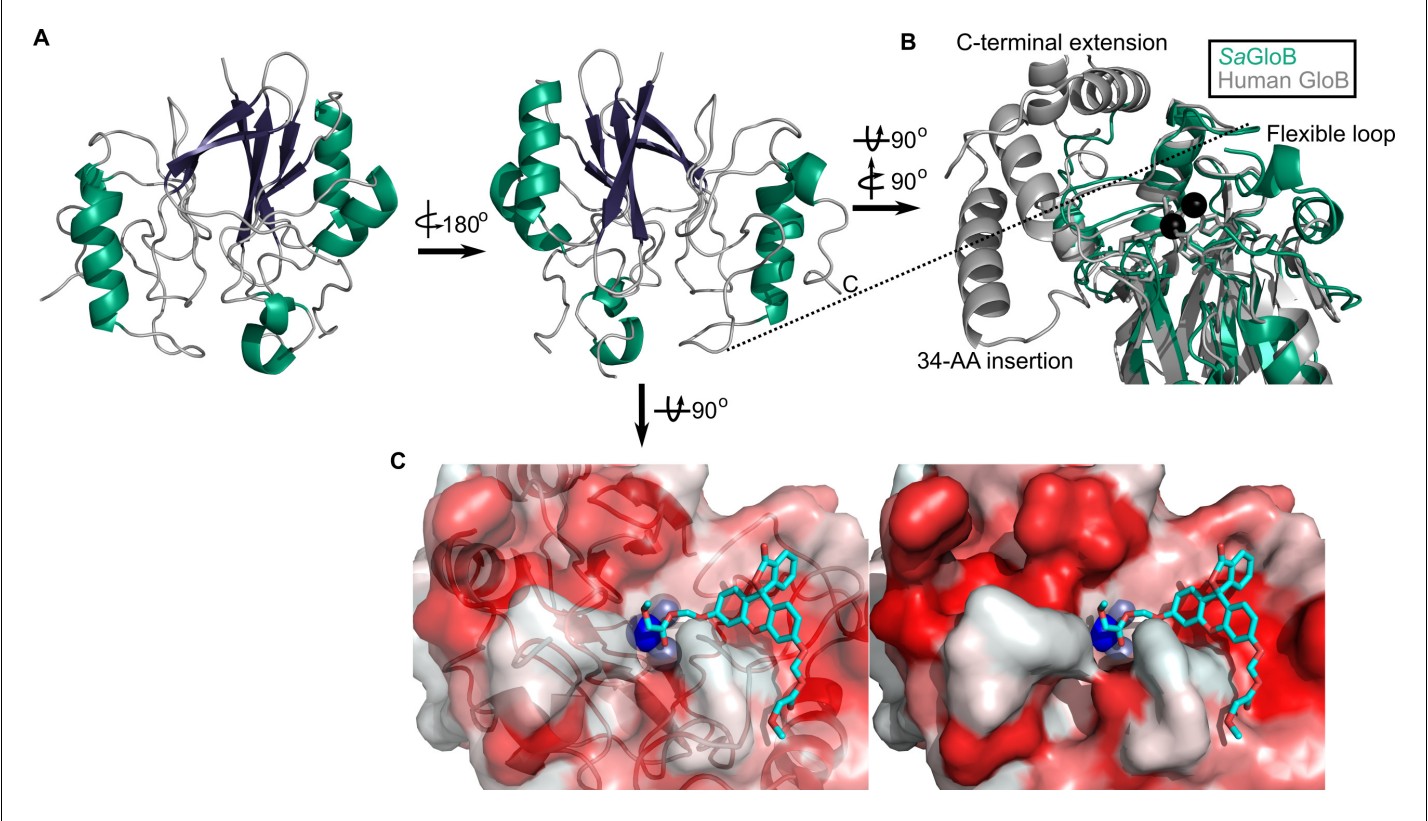

**Figure 5.** Three-dimensional structure of GloB. (**A**) Overall fold, a-alpha helices colored in green and β-strands colored in purple. (**B**) Comparison of SaGloB (green) and human GloB (gray). (**C**) Docking of the substrate 1O (sticks) in the active site of GloB. Left, partial cartoon view; right, surface view. White represents hydrophilic residues, whereas red represents hydrophobic residues. Zn ions indicated as silver spheres; water indicated as blue sphere.

The online version of this article includes the following source data and figure supplement(s) for figure 5:

**Source data 1.** Summary of crystallographic data collection and refinement statistics.
**Figure supplement 1.** Structural conservation of GloB.

pocket extends and opens significantly after passing by the oxyanion hole, supporting the capacity of FrmB to hydrolyze substrates that contain large steric groups distant from the carbonyl carbon, such as 11O.

## Three-dimensional structure of GloB

We also solved the structure of SaGloB to 1.65 Å, using selenomethionine (SeMet)-substituted GloB, and molecular replacement, using metallo-β-lactamase TTHA1623 from Thermus thermophilus as a search model (*Yamamura et al., 2009*). Final structural refinement parameters and statistics are summarized in *Figure 5—source data 1*. Four monomers of SaGloB are observed in the asymmetric unit with each displaying crystallographic symmetry. SaGloB exhibits the classic αβ/βα-fold that defines the metallo-β-lactamases, including glyoxalase II (*Figure 5A*; *Cameron et al., 1999*; *Honek, 2015*; *Stamp et al., 2010*; *Yamamura et al., 2009*).

As with SaFrmB, a DALI server search was performed to identify proteins structurally similar to SaGloB. SaGloB displays extremely high similarity to the unusual type II glyoxalase Ycbl from *Salmonella enterica* (PDB ID: 2XF4, r.m.s.d = 0.898 Å), to the molecular replacement search model TTHA1623 from *Thermus thermophilus* (PDB ID: 2ZWR, r.m.s.d. = 0.767), and to the *Arabidopsis thaliana* glyoxalase II (PDB ID: 1XM8, r.m.s.d. = 1.165 Å), with the exception that AtGloB has a 50 amino acid C-terminal extension (*Figure 5—figure supplement 1A*; *Marasinghe et al., 2005*; *Stamp et al., 2010*; *Yamamura et al., 2009*). Also consistent with previously observed GloB structures, SaGloB shows clear electron density for two zinc ions coordinated by six histidines and two

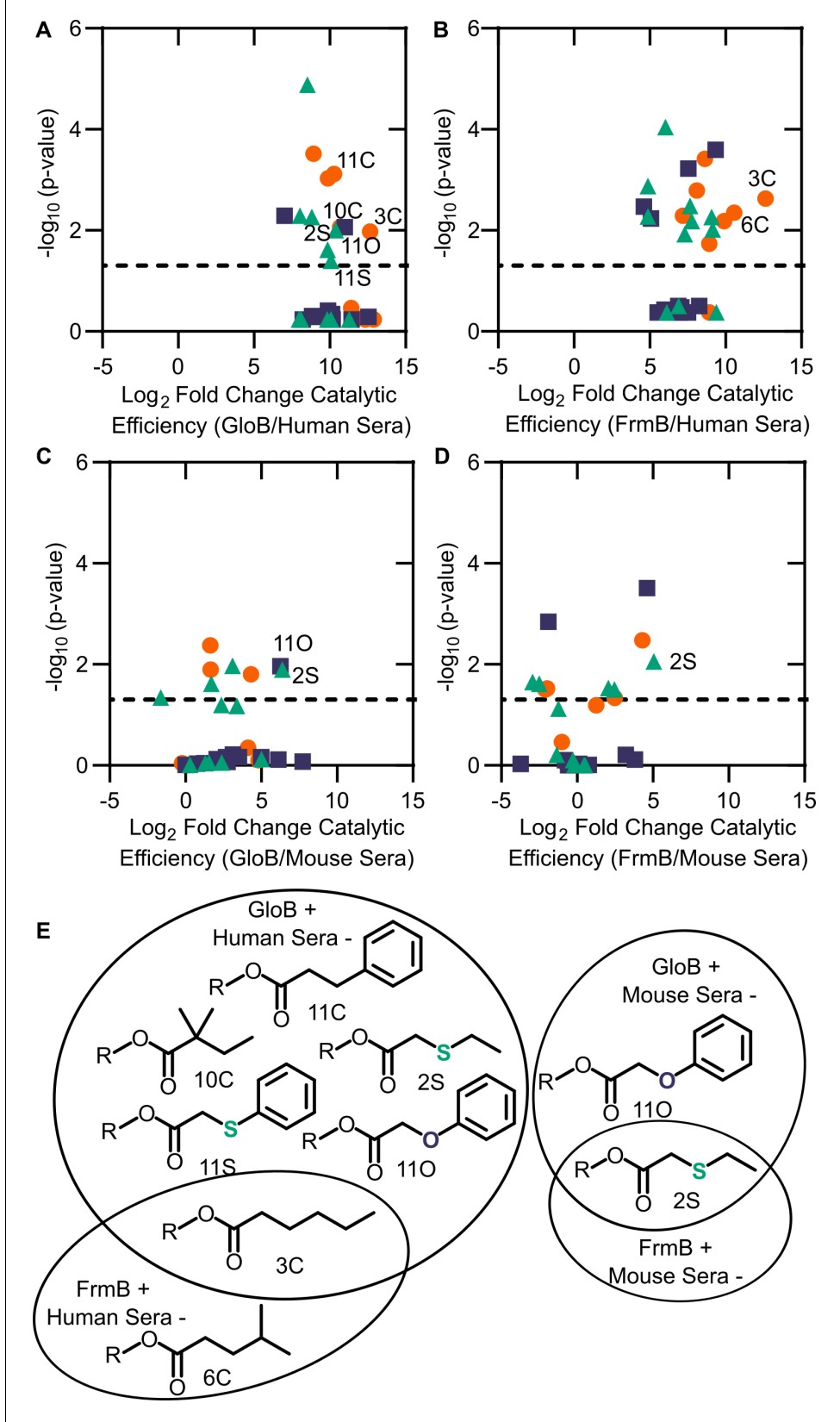

**Figure 6.** Comparison between microbial esterase and serum esterase catalytic efficiency. (**A–D**) Volcano plots of catalytic efficiency. Displayed are the means of three independent experiments. p-values calculated as pairwise
*Figure 6 continued on next page*

*Figure 6 continued*

t-tests with Holm-Sidak correction for multiple comparisons. (A) Comparison between human sera and GloB, (B) human sera and FrmB, (C) mouse sera and GloB, and (D) mouse sera and FrmB. (E) Structures of ester substrates with 210 enrichment in catalytic efficiency for microbial esterases over human serum (left) or 25 enrichment over mouse serum. Dashed line indicates a p-value of 0.05.

The online version of this article includes the following source data and figure supplement(s) for figure 6:

**Source data 1.** (1) Michaelis–Menten parameters for human sera.

**Figure supplement 1.** Comparison of esterase activity between fresh and lyophilized human sera.

**Figure supplement 2.** Modified catalytic efficiency (pmol fluorescein produced * $min^{-1}$*$\mu g^{-1}$ protein) of (A) human sera, (B) GloB, (C) FrmB, and (D) mouse sera.

**Figure supplement 3.** Comparison of mouse and human sera.

aspartates (*Figure 5—figure supplement 1B*). Density for a water molecule is also visible and appears to be coordinated by the two zinc ions, as observed for human glyoxalase II (*Cameron et al., 1999*).

Overlaying *S. aureus* GloB with *Homo sapiens* GloB (PDB: 1qh5) reveals that the two structures are remarkably similar (r.m.s.d = 1.249 Å), with a few notable exceptions. *Hs*GloB has two extensions: one, a 34-amino acid insertion; the other, a 32-amino acid C-terminal extension, both of which form α-helical features that abut the active site (*Figure 5B*; *Cameron et al., 1999*). On the opposite side of the active site, *Sa*GloB has a 19-amino acid flexible loop, which is partially observed in the electron density. This loop is positioned such that it may cover the active site to sterically hinder substrate access (*Figure 5B*). Overall, these structural differences between *Hs*GloB and *Sa*GloB support the design of prodrug substrates selectively cleaved by *S. aureus*.

We modeled the highest catalytic efficiency substrate for GloB, 1O, onto our structure. Autodock places 1O with its carbonyl oxygen adjacent to the active site water (*Figure 5C*; *Morris et al., 2009*; *Trott and Olson, 2010*). The GloB active site channel appears moderately wide, explaining why extensively branched substrates are not tolerated. Toward the end of the active site channel, GloB appears to form a tunnel. This tunnel is not reached by substrate 1O, but presumably would be

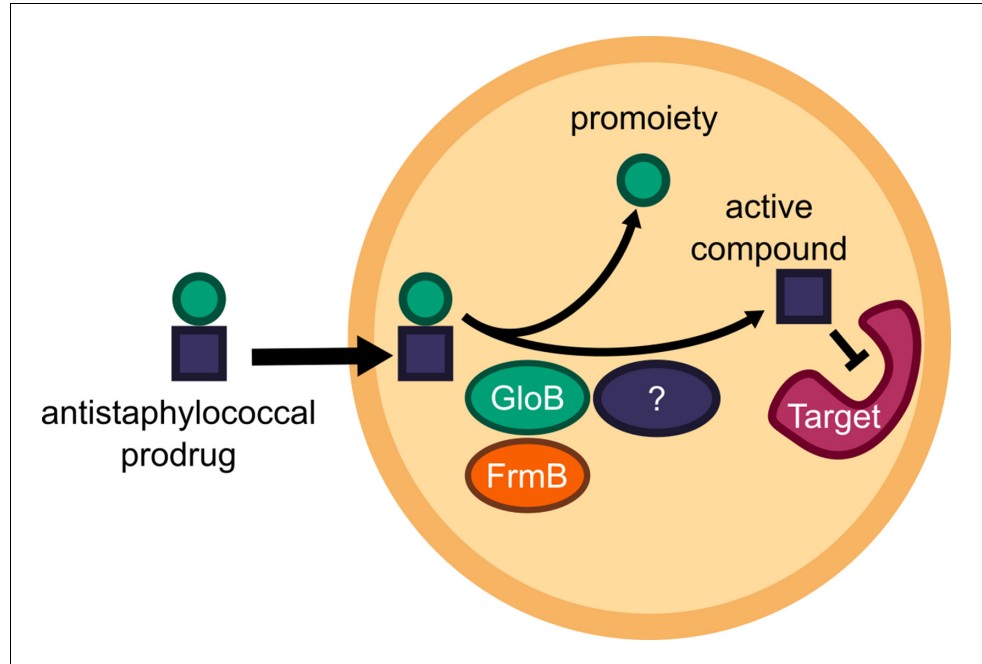

**Figure 7.** Model of antistaphylococcal prodrug activation. Lipophilic carboxy ester prodrugs transit the cell membrane, are first activated by either GloB or FrmB and at least one additional enzyme, before inhibiting the cellular target.

occupied in more sterically bulky substrates such as 11O. One arm of this tunnel is comprised of the highly flexible loop, which is only partially visible in our electron density, suggesting that during catalysis this flexible loop may accommodate larger substrates, such as 11O.

## Esterase specificity of human and mouse sera

We sought to evaluate whether ester promoieties could be designed for microbe-specific activation. Using the same 32-compound fluorescent substrate library, we determined each substrate's serum half-life. Both reconstituted and fresh sera functions are comparable in their activity and substrate preferences (*Figure 6—figure supplement 1*).

We performed full kinetic profiling of lyophilized human sera to define its ester substrate specificity. As sera is a mixture of multiple proteins instead of a single protein species, we report the $V_{max}/K_m$ normalized to the total amount of protein added to the assay (*Figure 6—figure supplement 2*). As observed for FrmB and GloB (*Figure 6—figure supplement 2A–C*), human sera has highest activity for oxygen and sulfur ether compounds. In contrast to FrmB and GloB, human sera is relatively uniform in its esterase activity across the substrate library. Short-chain substrates exhibit the highest catalytic efficiency, and though branching slightly reduces efficiency, it does not have as profound an impact as was observed with FrmB. The substrates displaying the poorest catalytic efficiency are universally the carbon series, in which added branching resulting in decreased substrate utilization.

As murine models are frequently used in the development and testing of novel pharmaceuticals, we also characterized the substrate preferences of mouse sera. Notably, mice are well known for their extremely active and broad serum esterase activity. Indeed, we find that mouse sera exhibit ~100-fold higher activity per mg serum protein than human sera (*Figure 6—figure supplements 2D* and *3*). This increase in activity is not uniform across the substrate library. Human sera is not as active on the carbon series and do comparatively better on the oxygen and sulfur ether compounds (*Figure 6—figure supplement 3*). Thus, our data suggest that mouse sera poorly predict human serum stability of ester prodrugs.

Finally, we sought to evaluate whether GloB and FrmB substrate specificities could be used for microbe-targeted prodrug design. As each esterase is likely to encounter multiple potential substrates in vivo, we utilized our modified $V_{max}/K_m$ as a comparator. We performed pairwise analysis for each combination of FrmB and GloB against human and mouse sera (*Figure 6A–D*). Using a cutoff of $2^{10}$-fold enrichment in activity for the microbial enzymes over the serum enzymes, FrmB displays a preference over human sera for two compounds – 3C and 6C, whereas GloB displays a preference for six compounds – 2S, 3C, 10C, 11C, 11O, and 11S (*Figure 6E*). Conversely, mouse sera hydrolyze all compounds within this cutoff. Lowering the cutoff to a $2^5$-fold enrichment in catalytic efficiency over mouse sera, FrmB and GloB are both more specific for compound 2S, and GloB additionally displays specificity for compound 11O.

## Discussion

Targeted microbial delivery and activation of lipophilic ester prodrugs is a highly desirable strategy to enable the expansion of druggable targets within bacteria while simultaneously improving drug selectivity. Identification of microbe-specific promoieties is crucial to this goal. Here, we identify two discrete esterases in *S. aureus*, FrmB and GloB, that activate the carboxy ester prodrug, POM-HEX (*Figure 7*). Sequence homologues of FrmB and GloB are found broadly within microbes, though substantial sequence variation exists. FrmB and GloB have distinct ester substrate specificities, which are supported by the structures of their active sites. Importantly, enzymatic substrate specificity correlates with the rate of ester activation in live bacterial cells. Accordingly, simple modifications to ester prodrugs are sufficient to change their rates of activation in vivo.

Simple ester modifications can also change the pattern of prodrug activation. For the successful development of microbially targeted ester prodrugs, compounds must resist human enzymes. Here we demonstrate that, although commonly considered 'nonspecific', human serum esterases exhibit specific ester substrate preferences. These preferences are distinct from mouse sera, suggesting that rodents are an inadequate model for ester prodrug activation, serum stability, and thus therapeutic potential of ester produgs in humans. We also find that the staphylococcal esterases, FrmB and GloB, utilize distinct patterns of substrates compared to the preferred substrates utilized by human sera, indicating the feasibility of developing microbially targeted ester prodrugs.

Further evaluation of these ester promoieties and their potential metabolic and pharmacological liabilities is necessary before extensive clinical application. For example, while human serum de-esterifies carboxy-ester compounds, the liver also contributes significantly to drug metabolism (*Laizure et al., 2013*; *Li et al., 2005*). Hepatocytes not only express human carboxylesterases 1 and 2, but also include a suite of cytochrome P450 enzymes that are important in facilitating drug clearance. Introduction of lipophilic promoieties such as those tested in this work also increases the logP, or hydrophobicity, of the drug. For highly charged phosphonates, such as POM-HEX, this increase in lipophilicity may be beneficial in improving intestinal absorption or cell penetration, as has been observed for the beta lactam cefditoren, which was formulated with a POM-prodrug moiety as cefditoren-pivoxyl to improve bioavailability (*Fuchs, 2007*). However, higher logP values raise concern for increased plasma protein binding and reduced efficacy in vivo. Whether increasing lipophilicity decreases the in vivo efficacy of these antibiotics remains to be tested, though decoration of the ester promoiety with more hydrophilic moieties is always possible.

Here, we have highlighted how *S. aureus* activates carboxy ester prodrugs. How microbes beyond *S. aureus* activate prodrugs, as well as the substrate specificities of pathogenic and commensal microbes, remains an important open question that will dictate the antimicrobial spectrum of a given ester prodrug. In past work, we have observed that lipophilic prodrugging with the POM-promoiety dramatically reduces compound activity in Gram-negative organisms (*Edwards et al., 2020*). The cause for this reduction remains unclear but may be explained by a lack of activating esterases in the tested Gram-negatives or an exclusion of the prodrugged molecule from the cell. Understanding this variability in compound activation will ultimately determine how selective targeted prodrug anti-infectives can be as well as their potential applications. Overall, this work paves the way for structure-guided development of *S. aureus*-specific prodrugs and establishes a pipeline for the identification of additional microbial prodrug-activating enzymes. We anticipate that these approaches will both guide the development of novel antimicrobials and lead to a more robust arsenal of anti-infective compounds with targeted specificity for the microbe over the human host.

# Materials and methods

**Key resources table**

| Reagent type (species) or resource | Designation | Source or reference | Identifiers | Additional information |
|---|---|---|---|---|
| Gene (*Staphylococcus aureus*) | GloB | GenBank | WP_001223008.1 | GloII |
| Gene (*Staphylococcus aureus*) | FrmB | GenBank | | estA |
| Strain, strain background (*Staphylococcus aureus*) | JE2 | BEI Resources | NR-46543 | |
| Strain, strain background (*Staphylococcus aureus*) | NE64 | Nebraska Transposon Mutant Library | Hypothetical protein | Available from BEI Resources as: NR-48501 |
| Strain, strain background (*Staphylococcus aureus*) | NE145 | Nebraska Transposon Mutant Library | protoporphyrinogen oxidase | Available from BEI Resources as: NR-48501 |
| Strain, strain background (*Staphylococcus aureus*) | NE202 | Nebraska Transposon Mutant Library | ABC transporter, ATP-binding protein, MsbA family | Available from BEI Resources as: NR-48501 |
| Strain, strain background (*Staphylococcus aureus*) | NE223 | Nebraska Transposon Mutant Library | hydroxyacylglutathione hydrolase | Available from BEI Resources as: NR-48501 |
| Strain, strain background (*Staphylococcus aureus*) | NE293 | Nebraska Transposon Mutant Library | staphylococcal accessory regulator Rot | Available from BEI Resources as: NR-48501 |
| Strain, strain background (*Staphylococcus aureus*) | NE355 | Nebraska Transposon Mutant Library | tributyrin esterase | Available from BEI Resources as: NR-48501 |
| Strain, strain background (*Staphylococcus aureus*) | NE364 | Nebraska Transposon Mutant Library | NAD-dependent epimerase/dehydratase family protein | Available from BEI Resources as: NR-48501 |

*Continued on next page*

*Continued*

| Reagent type (species) or resource | Designation | Source or reference | Identifiers | Additional information |
|---|---|---|---|---|
| Strain, strain background (*Staphylococcus aureus*) | NE377 | Nebraska Transposon Mutant Library | pyrroline-5-carboxylate reductase | Available from BEI Resources as: NR-48501 |
| Strain, strain background (*Staphylococcus aureus*) | NE386 | Nebraska Transposon Mutant Library | Hypothetical protein | Available from BEI Resources as: NR-48501 |
| Strain, strain background (*Staphylococcus aureus*) | NE478 | Nebraska Transposon Mutant Library | peptide ABC transporter, permease protein | Available from BEI Resources as: NR-48501 |
| Strain, strain background (*Staphylococcus aureus*) | NE503 | Nebraska Transposon Mutant Library | conserved hypothetical protein | Available from BEI Resources as: NR-48501 |
| Strain, strain background (*Staphylococcus aureus*) | NE520 | Nebraska Transposon Mutant Library | sensor histidine kinase SaeS | Available from BEI Resources as: NR-48501 |
| Strain, strain background (*Staphylococcus aureus*) | NE532 | Nebraska Transposon Mutant Library | PTS system, mannitol specific IIBC component | Available from BEI Resources as: NR-48501 |
| Strain, strain background (*Staphylococcus aureus*) | NE541 | Nebraska Transposon Mutant Library | alkaline phosphatase synthesis transcriptional regulatory protein | Available from BEI Resources as: NR-48501 |
| Strain, strain background (*Staphylococcus aureus*) | NE621 | Nebraska Transposon Mutant Library | Hypothetical protein | Available from BEI Resources as: NR-48501 |
| Strain, strain background (*Staphylococcus aureus*) | NE812 | Nebraska Transposon Mutant Library | tetrahydrodipicolinate acetyltransferase | Available from BEI Resources as: NR-48501 |
| Strain, strain background (*Staphylococcus aureus*) | NE874 | Nebraska Transposon Mutant Library | Hypothetical protein | Available from BEI Resources as: NR-48501 |
| Strain, strain background (*Staphylococcus aureus*) | NE929 | Nebraska Transposon Mutant Library | acetyl-CoA carboxylase, biotin carboxylase | Available from BEI Resources as: NR-48501 |
| Strain, strain background (*Staphylococcus aureus*) | NE937 | Nebraska Transposon Mutant Library | tandem lipoprotein | Available from BEI Resources as: NR-48501 |
| Strain, strain background (*Staphylococcus aureus*) | NE949 | Nebraska Transposon Mutant Library | Putative Transposase | Available from BEI Resources as: NR-48501 |
| Strain, strain background (*Staphylococcus aureus*) | NE1039 | Nebraska Transposon Mutant Library | excinuclease ABC, A subunit | Available from BEI Resources as: NR-48501 |
| Strain, strain background (*Staphylococcus aureus*) | NE1051 | Nebraska Transposon Mutant Library | Hypothetical protein | Available from BEI Resources as: NR-48501 |
| Strain, strain background (*Staphylococcus aureus*) | NE1071 | Nebraska Transposon Mutant Library | Hypothetical protein | Available from BEI Resources as: NR-48501 |
| Strain, strain background (*Staphylococcus aureus*) | NE1118 | Nebraska Transposon Mutant Library | dihydrolipoamide dehydrogenase | Available from BEI Resources as: NR-48501 |
| Strain, strain background (*Staphylococcus aureus*) | NE1127 | Nebraska Transposon Mutant Library | gamma-hemolysin component B | Available from BEI Resources as: NR-48501 |
| Strain, strain background (*Staphylococcus aureus*) | NE1173 | Nebraska Transposon Mutant Library | tRNA pseudouridine synthase A | Available from BEI Resources as: NR-48501 |
| Strain, strain background (*Staphylococcus aureus*) | NE1225 | Nebraska Transposon Mutant Library | ABC transporter, ATP-binding protein | Available from BEI Resources as: NR-48501 |
| Strain, strain background (*Staphylococcus aureus*) | NE1238 | Nebraska Transposon Mutant Library | transcriptional regulator, TetR family | Available from BEI Resources as: NR-48501 |
| Strain, strain background (*Staphylococcus aureus*) | NE1283 | Nebraska Transposon Mutant Library | Hypothetical protein | Available from BEI Resources as: NR-48501 |
| Strain, strain background (*Staphylococcus aureus*) | NE1296 | Nebraska Transposon Mutant Library | hydroxyacylglutathione hydrolase | Available from BEI Resources as: NR-48501 |
| Strain, strain background (*Staphylococcus aureus*) | NE1486 | Nebraska Transposon Mutant Library | phosphonate ABC transporter, permease protein | Available from BEI Resources as: NR-48501 |
| Strain, strain background (*Staphylococcus aureus*) | NE1505 | Nebraska Transposon Mutant Library | tributyrin esterase | Available from BEI Resources as: NR-48501 |

*Continued on next page*

*Continued*

| Reagent type (species) or resource | Designation | Source or reference | Identifiers | Additional information |
|---|---|---|---|---|
| Strain, strain background (*Staphylococcus aureus*) | NE1519 | Nebraska Transposon Mutant Library | Hypothetical Alkaline Phosphatase | Available from BEI Resources as: NR-48501 |
| Strain, strain background (*Staphylococcus aureus*) | NE1547 | Nebraska Transposon Mutant Library | Hypothetical protein | Available from BEI Resources as: NR-48501 |
| Strain, strain background (*Staphylococcus aureus*) | NE1610 | Nebraska Transposon Mutant Library | Hypothetical protein | Available from BEI Resources as: NR-48501 |
| Strain, strain background (*Staphylococcus aureus*) | NE1682 | Nebraska Transposon Mutant Library | PfkB family kinase | Available from BEI Resources as: NR-48501 |
| Strain, strain background (*Staphylococcus aureus*) | NE1723 | Nebraska Transposon Mutant Library | type I restriction-modification system, M subunit | Available from BEI Resources as: NR-48501 |
| Strain, strain background (*Escherichia coli*) | BL21 (DE3) | Sigma | CMC0014 | Chemically Competent |
| Biological sample (*Homo sapiens*) | Fresh Human Serum | This paper | | Whole blood was collected from a willing volunteer into untreated BD vacutaier tubes (BD, BD366430), allowed to clot, and aggregates were removed via centrifugation |
| Biological sample (*Homo sapiens*) | Lyophilized Human Serum | Rockland Inc | D314-5 | |
| Biological sample (*Mus musculus*) | Lyophilized Mouse Serum | Rockland Inc | D308-5 | |
| Recombinant DNA reagent | pET28a-*Sa*FrmB | This paper | | *E. coli* expression plasmid for *Sa*FrmB with cleavable HIS tag. |
| Recombinant DNA reagent | pET28a-*Sa*GloB | This paper | | *E. coli* expression plasmid for *Sa*GloB with cleavable HIS tag. |
| Recombinant DNA reagent | BG1861-*Sa*GloB | This paper | | *E. coli* expression plasmid for *Sa*GloB with HIS tag. |
| Recombinant DNA reagent | BG1861-*Sa*FrmB | This paper | | *E. coli* expression plasmid for *Sa*FrmB with HIS tag. |
| Sequence-based reagent | NWMN_0144_F | This paper | PCR Primer | TTTTCCTGATCCTGATTCAC |
| Sequence-based reagent | NWMN_0144_R | This paper | PCR Primer | ATGATGCTTCCATGTTTGTT |
| Sequence-based reagent | NWMN_0306_F | This paper | PCR Primer | AATACACCGGGTAACACAAC |
| Sequence-based reagent | NWMN_0306_R | This Paper | PCR Primer | CGTTTTGTTGAGCTAATTCC |
| Sequence-based reagent | NWMN_0309_F | This paper | PCR Primer | ACCATGCTTAAAGGGATTTT |
| Sequence-based reagent | NWMN_0309_R | This Paper | PCR Primer | TGTCACCTAAGTCAACACCA |
| Sequence-based reagent | NWMN_0407 (lpl4nm) _F | This paper | PCR Primer | CCGTTGGAGATAGGAAGTTA |
| Sequence-based reagent | NWMN_0407 (lpl4nm) _R | This paper | PCR Primer | TTTGTGCTTCTTTTGAACCT |
| Sequence-based reagent | NWMN_0654_F | This paper | PCR Primer | GAAAATGGAAGACTGATTGC |
| Sequence-based reagent | NWMN_0654_R | This paper | PCR Primer | TAATGCATCTGACAAAGTCG |
| Sequence-based reagent | NWMN_0762_F | This paper | PCR Primer | GGTGAAGTTTTGGACGATAA |

*Continued*

| Reagent type (species) or resource | Designation | Source or reference | Identifiers | Additional information |
|---|---|---|---|---|
| Sequence-based reagent | NWMN_0762_R | This paper | PCR Primer | TTTTCATCTGTCCGACTTTT |
| Sequence-based reagent | NWMN_1101_F | This paper | PCR Primer | TCCACCTATTGGAATTATCG |
| Sequence-based reagent | NWMN_1101_R | This paper | PCR Primer | AGACGTTCAATTTCAGTGCT |
| Sequence-based reagent | NWMN_1192 (pgsA) _F | This paper | PCR Primer | TGGGACGAAGTAATTACAGTT |
| Sequence-based reagent | NWMN_1192 (pgsA) _R | This paper | PCR Primer | ATATCCCCCTTGTATCGTTT |
| Sequence-based reagent | NWMN_1308 (dapD) _F | This paper | PCR Primer | TCTATTCGTGGAGGTACGAT |
| Sequence-based reagent | NWMN_1308 (dapD) _R | This paper | PCR Primer | ATCGTATGTGAGCCATTACC |
| Sequence-based reagent | NWMN_1410_F | This paper | PCR Primer | CGATAAACCTAAACCACTCG |
| Sequence-based reagent | NWMN_1410_R | This paper | PCR Primer | ATAAACAATGCTTGCCAAAT |
| Sequence-based reagent | NWMN_1505_F | This paper | PCR Primer | TGAAGGTGAATTAAGCGATG |
| Sequence-based reagent | NWMN_1505_R | This paper | PCR Primer | TGCTATTCCCAATTTGTTCA |
| Sequence-based reagent | NWMN_1655_F | This paper | PCR Primer | GAATTGTTGCAATTTAATGGT |
| Sequence-based reagent | NWMN_1655_R | This paper | PCR Primer | AACGTAATCATGCTCCATTC |
| Sequence-based reagent | NWMN_1679_F | This paper | PCR Primer | CCATGGGAAAAATTAGACAA |
| Sequence-based reagent | NWMN_1679_R | This paper | PCR Primer | AAATATCGCCTCACCTTTTT |
| Sequence-based reagent | NWMN_1723 (hemY) _F | This paper | PCR Primer | GCCGAATACACATCCATTAT |
| Sequence-based reagent | NWMN_1723 (hemY) _R | This paper | PCR Primer | AACCTTTGTCTCTGCTTCAA |
| Sequence-based reagent | NWMN_1851 (nadC) _F | This paper | PCR Primer | AGCCATTTTAGCACCATAAA |
| Sequence-based reagent | NWMN_1851 (nadC)_R | This paper | PCR Primer | TAGAATCCTGTCCTCCTGAA |
| Sequence-based reagent | NWMN_2057 (mtlF)_F | This paper | PCR Primer | TGTACAACGGTGTTGTTTTG |
| Sequence-based reagent | NWMN_2057 (mtlF)_R | This paper | PCR Primer | CGGTGAATAGTACGAGAGGA |
| Sequence-based reagent | NWMN_2528_F | This paper | PCR Primer | ACTGATGCTTTACCAGAAAC |
| Sequence-based reagent | NWMN_2528_R | This paper | PCR Primer | TCAGCGGTAGTAATAAAGGT |
| Chemical compound, drug | POM-HEX | *White et al., 2018* | | |
| Chemical compound, drug | Hemi-HEX | *Lin et al., 2020* | | |
| Chemical compound, drug | HEX | Lin et al. 2018 | | |

*Continued on next page*

*Continued*

| Reagent type (species) or resource | Designation | Source or reference | Identifiers | Additional information |
|---|---|---|---|---|
| Chemical compound, drug | Fluorescent Prosubstrates | *White et al., 2018* | | |
| Chemical compound, drug | S-D-lactoylglutathione | Sigma-Aldrich | L7140 | |
| Chemical compound, drug | 4-Nitrophenyl acetate | Sigma-Aldrich | N8130 | |
| Chemical compound, drug | 4-Nitrophenyl butyrate | Sigma-Aldrich | N9876 | |
| Chemical compound, drug | 4-Nitrophenyl trimethyl acetate | Sigma-Aldrich | 135046 | |
| Software, algorithm | WhatsGNU | *Moustafa and Planet, 2020* | | https://github.com/ahmedmagds/WhatsGNU |
| Software, algorithm | MUSCLE | *Letunic and Bork, 2019* | | https://www.ebi.ac.uk/Tools/msa/muscle/ |
| Software, algorithm | iTOL | *Madeira et al., 2019* | | https://itol.embl.de/ |
| Software, algorithm | HKL-3000 | *Minor et al., 2006* | | https://hkl-xray.com/hkl-3000 |
| Software, algorithm | PHASER | *McCoy et al., 2007* | | |
| Software, algorithm | COOT | *Emsley and Cowtan, 2004* | | https://www2.mrc-lmb.cam.ac.uk/personal/pemsley/coot/ |
| Software, algorithm | PHENIX | *Adams et al., 2010* | | http://www.phenix-online.org/ |
| Software, algorithm | ChemDraw3D | https://www.cambridgesoft.com/Ensemble_for_Chemistry/details/Default.aspx?fid=13&pid=668 | | |
| Software, algorithm | AutoDock Tools 1.5.7 | *Morris et al., 2009* | | http://autodock.scripps.edu/resources/adt |
| Software, algorithm | AutoDock Vina | *Trott and Olson, 2010* | | http://vina.scripps.edu/ |
| Software, algorithm | GraphPad Prism | https://www.graphpad.com/scientific-software/prism/ | | |
| Other | CellASIC ONIX2 microfluidic plate | EMD-Millipore | B04A-03-5PK | |

## Materials

POM-HEX, Hemi-HEX, and HEX were synthesized and resuspended in 100% DMSO as described previously (*Lin et al., 2020*). Fluorescent ester compounds were generously provided by the laboratory of Geoffrey Hoops (Butler University), and synthesis and characterization have been previously described (*White et al., 2018*). Pooled, delipidated, defibrinated, and lyophilized human and mouse serum was obtained from Rockland Inc.

## Quantification of resistance

Half-maximal inhibitory concentration ($IC_{50}$) determination was performed using microtiter broth dilution in clear 96-well plates (*European Committee for Antimicrobial Susceptibility Testing (EUCAST) of the ES of CM and ID (ESCMID), 2000*). Briefly, POM-HEX was added to 75 µL LB media at a final concentration of 20–50 µM POM-HEX and 0.5% DMSO, with POM-HEX concentrations varying according to resistant strain. Subsequently, POM-HEX was serially diluted in LB media containing 0.5% DMSO for a total of 10 dilutions. Two wells were left without drug, one used to define 100% growth, and the other used to control for media contamination and to define 0% growth. Seventy-five microliters of mid-log phase *S. aureus* diluted to $1 \times 10^5$ colony forming units/mL were subsequently added to the plate. Following inoculation, plates were incubated at 37°C with shaking, and $OD_{600}$ measurements were taken every 20 min for a total of 16 hr. Half-maximal inhibitory concentrations ($IC_{50}$) were determined by fitting the $OD_{600}$ of each condition following 10 hr of growth to a nonlinear regression using GraphPad Prism software. Experiments were performed in triplicate with technical duplicates.

Generation of POM-HEX resistant strains was performed by plating log-phase *S. aureus* Newman on LB agar containing 3.33 µM POM-HEX and incubating at 37°C overnight. Surviving single colonies were grown overnight in LB media and frozen in 10% glycerol for long term storage. All assays were performed from fresh overnight inoculations from glycerol stocks.

## Antibiotic resistance frequency

Antibiotic resistance frequency was determined on LB agar plates containing 4× and 10×, the minimum inhibitory concentrations of several antibiotics. MICs were determined for *S. aureus* Newman in liquid LB media. Antibiotics (MIC for *S. aureus* Newman) tested were as follows: Vancomycin (2 µg/mL), Clindamycin (0.25 µg/mL), Rifampicin (7.5 ng/mL), Nafcillin (0.5 µg/mL), Linezolid (1.67 µg/mL), and POM-HEX (3.13 µg/mL). Briefly, *S. aureus* Newman was grown in LB media to late log-phase and centrifuged at 4000 × g for 15 min before resuspension at an $OD_{600\ nm}$ of 30 (~1E11 CFU/mL) in LB media. One hundred microliter aliquots of concentrated *S. aureus* was plated in duplicate on selective and non-selective plates and incubated at 37°C. After 24 hr, mutation rates were calculated by counting the number of colonies present and dividing by the total number of colonies plated. Experiments were performed in triplicate.

## Whole-genome sequencing and variant discovery

Genomic DNA integrity was determined using Agilent 4200 Tapestation. Library preparation was performed with 0.25–0.5 µg of DNA. DNA was fragmented using a Covaris E220 sonicator using peak incident power 175, duty factor 10%, cycles per burst 200 for 240 s at 4°C. DNA was blunt ended, had an A base added to the 3' ends, and then had Illumina sequencing adapters ligated to the ends. Ligated fragments were then amplified for nine cycles using primers incorporating unique dual index tags. Fragments were sequenced on an Illumina MiSeq using paired-end reads extending 150 bases.

For the analysis, the WT-assembled genomic sequence for *S. aureus* Newman was retrieved from GenBank (accession number AP009351.1), and the paired-end reads were aligned to this genome using Novoalign v3.03 (Novocraft Technologies). Duplicates were removed and variants were called using SAMtools (*Li et al., 2009*). SNPs were filtered against the sequenced parental strain, and genetic variants were annotated using SnpEff v4.3 (*Cingolani et al., 2012*). Whole-genome sequencing data is available in the NCBI BioProject database and Sequence Read Archive under the BioProject ID PRJNA648156. All identified variants were verified via Sanger sequencing using the gene-specific primers found in the key resources table. Amplicons were sequenced by GeneWiz Inc (Bejing, China).

## WhatsGNU analysis

The *S. aureus* database was used to produce WhatsGNU proteomic reports for all the strains using WhatsGNU_main.py script in the ortholog mode. Eighteen *S. aureus* (nine atopic dermatitis [AD] and nine soft and skin tissue infection [SSTI]) isolates from an ongoing project representing different clonal complexes (CC1/5/8/22/30) were used for the comparison. The CC details for the 18 isolates are provided in *Figure 2—source data 4*. The reports were then used to produce a heat map of the GNU scores of GloB and FrmB using the heat map function in the WhatsGNU_plotter.py script. The heatmap was annotated with the ortholog variant rarity index where 'r' represents a rare GNU score (in the context of other alleles in the same protein ortholog group).

## Phylogenetic tree construction

Sequences of GloB, FrmB, and RpoB orthologs were retrieved from NCBI using the BlastP function with each organism on the tree as an individual search set. Of the returned sequences, the first complete sequence with the lowest E-value was selected for further analysis. Organisms were selected to include a wide variety of pathogenic and commensal microbes (*Yatsunenko et al., 2012*). In one instance, several of the top *E. coli* sequences were found to be highly similar to *S. aureus*, and on further analysis, we discovered that the original sequencing samples had high levels of *S. aureus* reads. These contaminated sequences were disregarded in our analysis. Sequence alignment was performed using MUSCLE using the default parameters, and the unrooted phylogenetic trees were visualized using iTOL (*Letunic and Bork, 2019*; *Madeira et al., 2019*).

## Recombinant expression and purification of FrmB and GloB

WT *frmB* and *gloB* sequences from *S. aureus* Newman were cloned into the BG1861 vector by Gene-Wiz Inc to introduce a hexahistidine tag (*Alexandrov et al., 2004*). The resultant plasmids were transformed into Stellar chemically competent cells (Clontech Laboratories), selected with carbenicillin, and the sequence was confirmed by Sanger sequencing. Subsequently, plasmids were transformed into chemically competent BL21 (DE3) cells and selected with 50 µg/mL ampicillin. Overnight liquid cultures were diluted 1:500 into LB media supplemented with ampicillin, grown shaking at 220 rpm to an $OD_{600}$ of 0.5–0.8 at 37°C, chilled to 16°C, and induced with 0.5 mM isopropyl β-D-1-thiogalactopyranoside (IPTG) for 16–20 hr. Cells were harvested by centrifugation at 6000 × g for 10 min at 4°C. The cell pellet was lysed by sonication in 50 mL lysis buffer containing 25 mM Tris–HCl (pH 7.5), 250 mM NaCl, 20 mM imidazole, 1 mM $MgCl_2$, 10% glycerol, and 200 µM phenylmethylsulfonyl fluoride (PMSF). Cell debris were removed by centrifugation twice at 20,000 × g for 20 min. The hexahistidine-tagged proteins were affinity purified from soluble lysate using nickel agarose beads (Gold Biotechnology). Bound protein was washed with 50 mL lysis buffer before elution using 5 mL of elution buffer containing 25 mM Tris–HCl pH 7.5, 250 mM NaCl, 300 mM imidazole, 1 mM $MgCl_2$, 10% glycerol. Affinity purified protein was further purified over a HiLoad 16/60 Superdex 200 gel filtration column (GE Healthsciences) using an AKTA Explorer. FPLC buffer contained 25 mM Tris–HCl pH 7.5, 250 mM NaCl, 1 mM $MgCl_2$, and 10% glycerol. Fractions containing >90% pure protein (evaluated by SDS–PAGE) were concentrated using an Amicon Ultra-15 centrifugal unit (EMD Millipore) and flash frozen in liquid nitrogen before storage at −80°C.

Protein used during crystallography experiments was generated via the same *frmB* and *gloB* sequences, but expression was performed from vector pET28a. *frmB* was cloned into the pET28a vector by GeneWiz Inc (Beijing, China), and *gloB* was cloned from the BG1861 vector using the forward primer 5′- dTGCTCGAGTGCGGCCGCTTAACCGTGTAAAAATGGATTT-3′ and the reverse primer 5′-dCGCGCGGCAGCCATATGATGAGGATTTCAAGCTTAACTTT-3′. The PCR product was cloned into pET28a digested with restriction enzymes NotI and NdeI using InFusion HD Cloning (Takara Bio). Both cloning strategies introduce a hexahistidine tag followed by a thrombin cleavage sequence. The pET28a plasmids encoding either frmB or gloB were transformed into chemically competent *E. coli* BL21(DE3) cells. Protein expression of FrmB proceeded as previously described, except FrmB containing cells were grown in Terrific Broth medium (24 g/L yeast extract, 20 g/L tryptone, 4 mL/L glycerol, 17 mM $KH_2PO_4$, 72 mM $K_2HPO_4$).

Selenomethionine (SeMet)-labeled GloB was prepared according to *Van Duyne et al., 1993* with minor modifications . Briefly, overnight cultures were grown in LB media, washed, and resuspended in M9 minimal media (per liter: 64 g $Na_2HPO_4$, 15 g $KH_2PO_4$, 2.5 g NaCl, and 5 g $NH_4Cl$) supplemented with 50 mg EDTA, 8 mg $FeCl_3$, 0.5 mg $ZnCl_2$, 0.1 mg $CuCl_2$, 0.1 mg $CoCl_2$, 0.1 mg $H_3BO_3$, 16 mg $MnCl_2$, 0.1 mg $Ni_2SO_4$, 0.1 mg molybdic acid, 0.5 mg riboflavin, 0.5 mg niacinamide, 0.5 mg pyridoxine monohydrate, and 0.5 mg thiamine per liter. Resuspended cultures were grown overnight. The following day, cultures were back diluted 1:50 and grown to an $OD_{600}$ of 0.5–0.8 at 37°C. Once at the appropriate $OD_{600\ nm}$, the following amino acids were added to the culture media at: 100 mg/L: lysine, phenylalanine, and threonine, 50 mg/L: isoleucine, leucine, and valine, 60 mg/L: SeMet. Cultures were grown for an additional 15 min at 37°C before cells were chilled to 16°C and induced with 0.5 mM IPTG for 16–20 hr.

Protein purification of FrmB- and SeMet-labeled GloB for crystallography proceeded as previously except following affinity purification the elution was dialyzed for 16–20 hr at 4°C with 20 U thrombin protease (GE Healthsciences) to remove the hexahistidine tag. Dialysis buffer contained 50 mM Tris pH 7.5, 50 mM NaCl, and 1 mM $MgCl_2$. Following dialysis, uncleaved protein, the hexahistidine tag, and thrombin were removed by flowing dialyzed protein over a benzamidine sepharose and nickel agarose bead column (GE Healthsciences). Column flow through was further purified over a HiLoad 16/60 Superdex-200 gel filtration column (GE Healthsciences) equilibrated with dialysis buffer. Protein was concentrated to 8–10 mg/mL in an Amicon Ultra-15 centrifugal unit and frozen at −80°C.

## Glyoxalase II activity assay

Glyoxalase II activity was assessed as previously with minor changes (*Mikati et al., 2020*; *Stamp et al., 2010*). Briefly, reactions were mixed to form a final concentration of 25 mM Tris pH 7.5, 250 mM NaCl, 1 mM $MnCl_2$, 10% glycerol, 200 µM 5,5′-dithiobis(2-nitrobenzoic acid) (DTNB,

Sigma D8130), 1 mM D-lactoylglutathione (Sigma L7140), and 0.15–0.63 µg protein (130–550 nM GloB, 100–430 nM FrmB). Protein concentrations were varied to ensure the reaction was linear across protein concentrations. Reactions without D-lactoylglutathione were pre-incubated at 37°C for 10 min prior to assay initiation with the addition of substrate. Release of glutathione from D-lactoyl-glutathione was quantified spectrophotometrically at 37°C and 412 nm through the conversion of DTNB to TNB. Experiments were performed in triplicate with technical duplicates.

## 4-Nitrophenyl ester substrate activity assays

4-Nitrophenyl substrate-specific activity was determined in 50 µL reactions containing 25 mM Tris pH 7.5, 250 mM NaCl, 1 mM MnCl$_2$, 10% glycerol, 1 µM protein, and 1 mM 4-nitrophenyl substrate. The tested substrates, 4-nitrophenyl acetate (Sigma, N8130), 4-nitrophenyl butyrate (Sigma N9876), and 4-nitrophenyl trimethylacetate (Sigma 135046), were resuspended in acetonitrile at 100 mM. Reactions without 4-nitrophenyl substrate were preincubated at 37°C for 10 min prior to assay initiation via substrate addition. Conversion of 4-nitrophenyl substrates to 4-nitrophenol was tracked photometrically at 37°C and A$_{405nm}$. Experiments were performed in triplicate with technical duplicates.

## NMR characterization of GloB and FrmB activation products

Two hundred or 400 µM POM-HEX was incubated with 4 nmol GloB, FrmB, or 4 nmol each GloB and FrmB in 500 µL reactions. Reactions were buffered to a final concentration of 50 mM Tris pH 7.5, 50 mM NaCl, 1 mM MgCl$_2$. Reactions were allowed to proceed for 1 hr at 37°C prior to analysis. Samples were prepared for NMR studies by resuspending them in water and 10% (50 µL) D$_2$O (deuterium oxide 99.9% D, contains 0.75 wt% 3-(trimethylsilyl)propionic-2,2,3,3-d$_4$ acid, sodium salt, Sigma–Aldrich). NMR spectra are acquired on a Bruker Avance III HD 500 MHz spectrometer equipped with a cryoprobe. Two-dimensional (2D) $^1$H-$^{31}$P HSQC measurements were obtained using hsqcetgp pulse program (with duration of 15 min and scan parameters of two scans, td = 1024 and 256, gpz2 % = 32.40, $^{31}$P SW = 40 ppm, O2p = 20 ppm, cnst2 = 22.95) and analyzed using 3.1 Top-Spin. The 1D projection of columns excluding the water signal was obtained from the 2D $^1$H-$^{31}$P HSQC spectrum by obtaining spectra of positive projection of columns 1–600 and 650–1024 and adding them.

## Esterase substrate specificity determination using fluorogenic SAR library

Kinetic measurements were performed according to *White et al., 2018* with minor variation. Lyophilized human and mouse sera were resuspended according to manufacturer instructions in highly pure, filtered water at protein concentrations of 85 mg/mL and 70 mg/mL, respectively. One milliliter of resuspended serum was added to a 24 mL mastermix for a final concentration of 31.25 mM Tris pH 7.5, 312.5 mM NaCl, 1.25 mM MgCl$_2$, 12.5% glycerol, and 3.4 mg/mL or 2.8 mg/mL protein for human and mouse serum, respectively. For purified proteins, 5 mL of a 75 µg/mL stock was added to yield a 20 mL mastermix containing 31.25 mM Tris pH 7.5, 312.5 mM NaCl, 1.25 mM MgCl$_2$, 12.5% glycerol, and 18.75 µg/mL protein. Mastermix was stored on ice when not in use. Twenty microliters of mastermix was transferred to a black, 96-well half area microplate (Corning, CLS3993) and prewarmed at 37°C. Fluorogenic substrates were prepared as 10 mM stock solutions in 100% DMSO and were diluted in water to a starting concentration of 500 µM. Enzyme catalyzed substrate hydrolysis was initiated by addition of 5 µL substrate dilution in technical duplicate to the prewarmed serum or protein solution. Final assay concentrations were as follows: 25 mM Tris pH 7.5, 250 mM NaCl, 1 mM MgCl$_2$, 10% glycerol, and protein at a concentration of 2.72 mg/mL (human serum), 2.24 mg/mL (mouse serum), or 15 µg/mL (FrmB, GloB). The resulting change in fluorescence ($\lambda_{ex}$ = 485 nm, $\lambda_{em}$ = 520 nm) was followed for 15 min at 37°C, collecting data every 30 s on a FLUOstar Omega microplate reader (BMG Labtech). Fluorescence measurements were converted to molar concentrations using a fluorescein standard curve (2.5 nmol–0.6 pmol). The initial rates of reaction were measured three independent times with two technical replicates per measurement and fit to a line using GraphPad Prism (GraphPad Software, La Jolla, CA). Initial rates of reaction were plotted versus the concentration of substrate and fit to a standard Michaelis–Menten equation ($v = V_{max}[S]/(K_m + [S])$), yielding estimates of $V_{max}$ and $K_m$. Values for $k_{cat}$ and $k_{cat}/K_m$ were calculated based on amount of enzyme added when purified enzymes were used. For substrates

that did not display saturation in the velocity versus substrate experiment (i.e., $K_m >>> [S]$), $V_{max}/K_m$ was estimated based on v/[S].

## Fresh human serum

Fresh human serum was collected from a willing volunteer in untreated BD vacutainer tubes (BD, BD366430) (Washington University IRB # 201012782). Whole blood was allowed to clot at room temperature, and aggregates were separated from the remaining serum through centrifugation at 400 × g for 8 min. Sera were obtained from the same volunteer on two separate occasions.

## Serum half-life determination

Lyophilized human sera was obtained from Rockland Inc and resuspended in pure water. Twenty microliters lyophilized sera or fresh sera was prewarmed at 37°C in a 96-well half-area microplate (Corning, CLS3993). Following plate warming, 5 µL of the fluorogenic substrates were added to the plate for a final concentration of 25 µM. Substrate hydrolysis was tracked over a period of three h at 37°C, with fluorescence measurements ($\lambda_{ex}$ = 485 nm, $\lambda_{em}$ = 520 nm) being taken every two min on a FLUOstar Omega microplate reader (BMG Labtech). The resulting fluorescence values were converted to % substrate hydrolyzed using a fluorescein standard curve and fit to a one-phase decay model using GraphPad Prism. Experiments were performed in technical and biological duplicate.

## Microfluidics measurements on *S. aureus*

Overnight cultures of *S. aureus* were grown in LB media, back diluted 1:500, and grown to early exponential phase ($OD_{600}$ 0.1–0.15), then washed in 1× PBS, and loaded on a bacterial CellASIC ONIX2 microfluidic plate. Prior to cell loading, the microfluidics plate lines were flushed with 1× PBS + 1% DMSO or 10 µM fluorescent pro-substrate in 1× PBS + 1% DMSO, and the plate was preincubated at 37°C. The microfluidics plate was loaded onto a Nikon Ti-E inverted microscope (Nikon Instruments, Inc) equipped with a 100× Plan N (N.A. = 1.45) Ph3 objective, SOLA SE Light Engine (Lumencor), heated control chamber (OKO Labs), and an ORCA-Flash4.0 sCMOS camera (Hammamatsu Photonics). The GFP filter set was purchased from Chroma Technology Corporation. Cells were loaded into the chamber until a single field of view contained 50–150 cells or cell clusters. Following cell loading, 1× PBS was slowly flown through the flow cell (t = 0), and cells were observed in both phase and fluorescence microscopy for 10 min before the flow media was rapidly switched over a period of two min to 1× PBS containing 1% DMSO and 10 µM fluorescent prosubstrate before slow flow for the remainder of the experiment. Images were acquired every two min for a total of 44 min, and all experiments were undertaken at 37°C. The phase-contrast exposure time was kept constant at 200 ms, and the fluorescent channel exposure time was kept constant at 500 ms. For fluorescent images, the gain remained constant across all experiments. Image capture and analysis was performed using Nikon Elements Advanced Research software. Individual cells or clusters of cells were auto detected in the fluorescent channel using the intrinsic background fluorescence of each cell. Manual curation followed autodetection to remove debris or cells that did not stay within the field of view throughout the experiment. Fluorescent intensity for each individual cell or cluster of cells was measured through the duration of the experiment and normalized to the area of the identified cell to yield the mean fluorescent intensity. Background cell autofluorescence was corrected by subtracting the average fluorescence across all identified objects from t = 0 through t = 10. Each experiment was performed in duplicate, with >50 individual cells or clusters analyzed in each experiment.

## Protein crystallography, phasing, and data refinement

Crystals of *S. aureus* FrmB were grown at 16°C using vapor diffusion in 20 µL hanging drops containing a 1:1 mixture of protein (6 mg/mL) and crystallization buffer (0.1 M Tricine pH 7.7, 15% PEG6K, 2.5 M NaCl, 0.125% *n*-dodecyl-β-D-glucoside). Prior to data collection, crystals were stabilized in cryoprotectant (mother liquor supplemented with 20% glycerol) before flash freezing in liquid nitrogen for data collection at 100 K. Crystals of Se-Met-labeled *S. aureus* GloB were grown at 16°C using vapor diffusion in 2 µL hanging drops containing a 1:1 mixture of protein (8 mg/mL) and crystallization buffer (0.1 M imidazole pH 6.9, 0.2 M ammonium sulfate, 0.1 M calcium chloride, and 21% PEG

8 k). SeMet-labeled GloB crystals were stabilized in well solution supplemented with 15% glycerol and flash frozen in liquid nitrogen. All diffraction images were collected at beamline 19-ID of the Argonne National Laboratory Advanced Photon Source at Argonne National Laboratory. HKL3000 was used to index, integrate, and scale the data sets (*Minor et al., 2006*). To phase the initial dataset of FrmB, molecular replacement was performed in PHASER using the x-ray crystal structure of a low-temperature active alkaline esterase (PDB ID: 4RGY) as a search model (*Hu et al., 2015*; *McCoy et al., 2007*). SeMet-labeled GloB was phased using the x-ray crystal structure of TTHA1623 from *Thermus thermophilus* HB8 (PDB ID: 2ZWR) (*Yamamura et al., 2009*). Buccaneer was used to build both initial models, and subsequent, iterative rounds of model building and refinement used COOT and PHENIX, respectively (*Adams et al., 2010*; *Cowtan, 2006*; *Emsley and Cowtan, 2004*). Data collection and refinement statistics are summarized in *Figure 4—source data 1* and *Figure 5—source data 1*. Atomic coordinates and structure factors of *S.* aureus FrmB (PDB: 7L0A) and *S. aureus* GloB (PDB: 7L0B) are deposited in the RCSB Protein Data Bank.

## Substrate docking

GloB and FrmB structures were prepared for substrate autodocking using AutoDock Tools 1.5.7 (*Morris et al., 2009*). Metals and water molecules were removed from the crystal structure of FrmB as canonical serine hydrolases do not utilize metal in their reaction mechanism. Solvent water in the GloB crystal structure was removed, but the active site water and heavily coordinated zinc molecules were left in place. The three-dimensional structure of substrate 1O was generated using Chem-Draw3D and prepared for docking using AutoDock Tools 1.5.7. Substrate docking of FrmB and GloB was performed using AutoDock Vina (*Trott and Olson, 2010*).

## Acknowledgements

We are grateful to the students of the Spring 2020 Biol 4522 course at Washington University for the creation of FrmB point mutation plasmids and the purification of FrmB point mutants. Thank you to Vandna Kukshal and Jason Schaffer for helpful discussions around data and mechanisms, to the Bubeck-Wardenburg laboratory for sharing benchspace during revisions, and to Petra Levin for assistance with microfluidics and microscopy. AOJ is supported by NIH/NIAID R01-AI103280, R21-AI123808, and R21-AI130584, and AOJ is an Investigator in the Pathogenesis of Infectious Diseases (PATH) of the Burroughs Wellcome Fund. This publication was made possible in part by Grant Number UL1 RR024992 from the NIH-National Center for Research Resources (NCRR). RJJ was supported by a grant from the National Institutes of Health (R15-GM110641).

## Additional information

### Funding

| Funder | Grant reference number | Author |
| --- | --- | --- |
| National Institutes of Health | R01 AI103280 | Audrey R Odom John |
| National Institutes of Health | R21 AI123808 | Audrey R Odom John |
| National Institutes of Health | R21 AI130584 | Audrey R Odom John |
| National Center for Research Resources | UL1 RR024992 | Joseph M Jez |
| Burroughs Wellcome Fund | PATH | Audrey R Odom John |
| National Institutes of Health | R15-GM110641 | R Jeremy Johnson |

The funders had no role in study design, data collection and interpretation, or the decision to submit the work for publication.

### Author contributions

Justin J Miller, Conceptualization, Formal analysis, Investigation, Visualization, Methodology, Writing - original draft, Project administration, Writing - review and editing; Ishaan T Shah, Jayda Hatten,

Yasaman Barekatain, Elizabeth A Mueller, Ahmed M Moustafa, Formal analysis, Methodology, Writing - review and editing; Rachel L Edwards, Conceptualization, Writing - review and editing; Cynthia S Dowd, Conceptualization, Funding acquisition, Writing - review and editing; Geoffrey C Hoops, R Jeremy Johnson, Methodology; Paul J Planet, Formal analysis, Supervision, Writing - review and editing; Florian L Muller, Conceptualization, Supervision, Funding acquisition, Writing - review and editing; Joseph M Jez, Conceptualization, Supervision, Funding acquisition, Writing - original draft, Project administration; Audrey R Odom John, Conceptualization, Resources, Formal analysis, Supervision, Funding acquisition, Writing - original draft, Project administration, Writing - review and editing

### Author ORCIDs
Justin J Miller (iD) https://orcid.org/0000-0001-9400-8916
Elizabeth A Mueller (iD) http://orcid.org/0000-0001-5482-6551
Ahmed M Moustafa (iD) http://orcid.org/0000-0002-9949-6936
Audrey R Odom John (iD) https://orcid.org/0000-0001-8395-8537

### Decision letter and Author response
Decision letter https://doi.org/10.7554/eLife.66657.sa1
Author response https://doi.org/10.7554/eLife.66657.sa2

## Additional files

### Supplementary files
• Supplementary file 1. Primers used during this study.

• Transparent reporting form

### Data availability

Sequencing data and structure data is provided in the manuscript and deposited on the NCBI Bio-Project database (PRJNA648156) or in the protein data bank (FrmB accession code 7L0A, GloB accession code 7L0B).

The following dataset was generated:

| Author(s) | Year | Dataset title | Dataset URL | Database and Identifier |
|---|---|---|---|---|
| John O | 2021 | Understanding *Staphylococcus aureus* resistance to POM-HEX | https://www.ncbi.nlm.nih.gov/bioproject/?term=PRJNA648156 | NCBI BioProject, PRJNA648156 |

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
