## [Decision Letter]

**Acceptance summary:**

This submission reports an approach to identify bacterial specific carboxyesterases that can be exploited to activate prodrugs of antibiotics in *Staphylococcus aureus*. The premise is that charged functional groups found in some antibiotics prevent entry of these into bacteria, an effect that can be circumvented through esterification of the antibiotic to allow entry. Thereafter, activation of the antibiotic will occur in the bacterial cytoplasm through hydrolysis by bacterial esterases. The study reports an interesting and generalisable approach to discovery of prodrug-activating enzymes for use in developing new treatments for microbes.

**Decision letter after peer review:**

Thank you for submitting your article "Structure-guided microbial targeting of anti-staphylococcal prodrugs" for consideration by *eLife*. Your article has been reviewed by 2 peer reviewers, and the evaluation has been overseen by Bavesh Kana as the Senior and Reviewing Editor. The reviewers have opted to remain anonymous.

Essential revisions:

1. The issue of resistance is not addressed sufficiently. This strategy relies on the necessity of non-essential bacterial enzymes for drug activation, which be circumvented by bacteria through mutation of the drug activating enzymes. The dependence on microbial activation is always a liability as has been shown for *M. tuberculosis* agents such as isoniazid and pyrazinamide. Ultimately this can be tested experimentally, for example determining the resistant frequencies for HEX or another prodrug and comparing to frequencies for known antibiotics. This will strengthen the work and provide a more balanced view.

2. While serum esterase would clearly be a concern, there are also abundant esterases in the liver. No mention was made of stability using liver microsomes or, even better, hepatocytes to get a picture of overall metabolic stability of the leading pro-moieties.

3. There is little data on the effect of the ether group of the ester on activity, as experiments were done using a fluorescence based system. Some evidence that these selectivities extend to more real-world systems/antibiotics would be helpful

*Reviewer #1 (Recommendations for the authors):*

Overall, the manuscript is well written and is largely free of typical typographic errors or poorly rendered figures. The writing is clear and logically presented and was overall an enjoyable read. The work conducted appears to be of very high quality and would likely be of wide interest. I am impressed with the overall rigor of the work and have no major issues with the work that has been done. I do have some concerns that should be addressed that relate to the overall applicability and impact of this work.

(1) A major concern is resistance. This strategy relies on the necessity of non-essential enzymes for drug activation which presents a very simple and obvious mechanism of resistance. In fact, the studies themselves using HEX shows just how easy this can be. The dependence on microbial activation is always a liability as has been shown in *M. tuberculosis* agents such as isoniazid and pyrazinamide. Ultimately this can be tested experimentally, for example determining the resistant frequencies for HEX or another prodrug and comparing to frequencies for known antibiotics.

(2) While serum esterase would clearly be a concern, there are also abundant esterases in the liver. No mention was made of stability using liver microsomes or, even better, hepatocytes to get a picture of overall metabolic stability of the leading pro-moieties.

(3) There is little data on the effect of the ether group of the ester on activity as all were done using a fluorescene based system. It would be helpful to see some evidence that these selectivities extend to more real-world systems.

(4) It would be valuable to give an idea of how many opportunities there are for this and the spectrum of application. How many drug candidates are these for this? Permeability into Gram-positive organisms such as *S. aureus* is usually not much of an issue like it is in Gram-negatives. Outside of a few phosphonates, there aren't many that come to mind. This is also true of organisms, how broad-spectrum would one of these agents be? Perhaps the authors could show the activity of HEX against several pathogens.

*Reviewer #2 (Recommendations for the authors):*

Antimicrobial resistance is a serious problem and S. aureus infections are also still a serious public health threat. The authors presented an interesting approach to the use of prodrugs against *S. aureus* infections. The research paper is very solidly based on scientific results. The research behind this article is very extensive and the basic research presented is very interesting and offers the possibility of developing new antibiotics. I will give more of a perspective as a researcher from antibacterial drug discovery. The advantage of the study is that both crystal structures are also available, allowing structure-based design of antibiotics that could be hydrolyzed by the proposed enzymes.

– This work lacks a proof-of-concept to confirm the use of the proposed ester groups on real antibiotic molecules and to confirm hydrolysis in *S. aureus* by the two proposed enzymes FrmB and GloB.

– It is not clear for which functional groups this approach would be most appropriate, hydroxyl, carboxyl?

– With the introduction of ester groups, the logP is increased and so are inappropriate physicochemical properties. Due to the increased logP, plasma protein binding could be high, thus reducing in vivo efficacy.

– The last comment refers to the proposed approach for Gram-positive bacteria, where crossing antibiotics through the bacterial cell wall is usually not so problematic. The approach would be much more interesting and necessary for Gram-negative bacteria.

---

## [Author Response]

Essential revisions:1. The issue of resistance is not addressed sufficiently. This strategy relies on the necessity of non-essential bacterial enzymes for drug activation, which be circumvented by bacteria through mutation of the drug activating enzymes. The dependence on microbial activation is always a liability as has been shown for M. tuberculosis agents such as isoniazid and pyrazinamide. Ultimately this can be tested experimentally, for example determining the resistant frequencies for HEX or another prodrug and comparing to frequencies for known antibiotics. This will strengthen the work and provide a more balanced view.

The reviewers raise an important point regarding the likelihood of resistance based on loss-offunction of GloB and FrmB. Although GloB and FrmB are dispensable in vitro in rich media, this may not be the case in ecological niches (such as infection) where nutrients are more limited and/or additional selective pressures exist. Our findings that GloB and FrmB have extremely high sequence conservation across *S. aureus* isolates (Figure 2 supplement 1) suggests that both proteins have an important biological function that is not immediately evident in rich media. Further, we envision this manuscript as a generalizable approach to discovery of prodrug-activating enzymes for use in developing microbially targeted prodrugs (relatively unexplored in the current literature).

To experimentally address the reviewers’ concerns, we have also determined the resistance frequency of *S. aureus* Newman to our novel inhibitor, POM-HEX, as well as to clinically relevant antimicrobials including the following: vancomycin, clindamycin, rifampicin, nafcillin, and linezolid at 4x and 10x the MIC of each antibiotic respectively. These new data are included in our manuscript. These results reveal that spontaneous resistance to POM-HEX occurs in rich media somewhat more frequently than rifampicin, and is now included as a new Figure 2, supplement 3 and discussed in the results (lines 166-177).

2. While serum esterase would clearly be a concern, there are also abundant esterases in the liver. No mention was made of stability using liver microsomes or, even better, hepatocytes to get a picture of overall metabolic stability of the leading pro-moieties.

We thank the reviewers for this feedback and agree that evaluating promoiety stability with regards to liver metabolism is also important. Ongoing work in the lab specifically addresses the role of hepatocytes and liver microsomes, to extend our characterization of esterase specificity to multiple cell types and additional pathogens. We have added a section to the discussion to include this important caveat (lines 461-474).

3. There is little data on the effect of the ether group of the ester on activity, as experiments were done using a fluorescence based system. Some evidence that these selectivities extend to more real-world systems/antibiotics would be helpful

As the reviewers point out, our substrate specificity studies specifically address the structure-activity relationship of which prodrug moieties are best recognized by serum and the staphylococcal esterases GloB and FrmB. Our prior work on staphylococcal GloB indicates that GloB homologs de-esterify POM promoieties from structurally diverse antibacterial “warheads” (PMID: 33118347, including the FDA-approved cefditoren pivoxil, see Figure 4 of that manuscript), and our ongoing studies address the degree to which this is true for FrmB and mammalian serum esterases. We believe that our current studies provide novelty in the use of comparative serum vs. microbial esterase specificity to facilitate design of selective esters that may be useful for selective pathogen targeting, irrespective of the warhead.

Reviewer #1 (Recommendations for the authors):Overall, the manuscript is well written and is largely free of typical typographic errors or poorly rendered figures. The writing is clear and logically presented and was overall an enjoyable read. The work conducted appears to be of very high quality and would likely be of wide interest. I am impressed with the overall rigor of the work and have no major issues with the work that has been done. I do have some concerns that should be addressed that relate to the overall applicability and impact of this work.1) A major concern is resistance. This strategy relies on the necessity of non-essential enzymes for drug activation which presents a very simple and obvious mechanism of resistance. In fact, the studies themselves using HEX shows just how easy this can be. The dependence on microbial activation is always a liability as has been shown in M. tuberculosis agents such as isoniazid and pyrazinamide. Ultimately this can be tested experimentally, for example determining the resistant frequencies for HEX or another prodrug and comparing to frequencies for known antibiotics.

We thank the reviewer for their thoughtful and thorough review of our work. We agree that resistance via GloB and FrmB will be a concern for the development of prodrugs utilizing these enzymes for activation. However, while GloB and FrmB are dispensable in vitro in rich media, this may not be the case during an infection setting where nutrients are more limited and additional selective pressures exist. Our finding that GloB and FrmB have extremely high sequence conservation across sequenced *S. aureus* strains (Figure 2 supplement 1) suggests that both proteins have an important biological function that is not uncovered in rich media. Further, we envision this manuscript as a generalizable approach to discovery of prodrug-activating enzymes for use in developing microbially targeted prodrugs. prodrug activation can be targeted (relatively unexplored in the literature currently). With the former in mind, it may prove valuable to unearth essential esterases and perform extensive characterization as we have done here for GloB and FrmB.

To experimentally address the reviewers’ concerns, we have additionally determined the resistance frequency of *S. aureus* Newman to POM-HEX as well as to clinically relevant antimicrobials including the following: vancomycin, clindamycin, rifampicin, nafcillin, and linezolid at 4x and 10x the MIC of each antibiotic respectively and included these findings in our manuscript. These results reveal that spontaneous resistance to POM-HEX occurs in rich media at somewhat frequency than rifampicin, and is now included as a new Figure 2, supplement 3 and discussed in the results (lines 166-177).

2) While serum esterase would clearly be a concern, there are also abundant esterases in the liver. No mention was made of stability using liver microsomes or, even better, hepatocytes to get a picture of overall metabolic stability of the leading pro-moieties.

We thank the reviewers for this feedback and agree that evaluating promoiety stability in liver systems is critically important. Ongoing work in the lab specifically addresses the role of hepatocytes and liver microsomes, to extend our characterization of esterase specificity to multiple cell types and additional pathogens. We have added a section to the discussion to include this caveat (lines 461-474).

3) There is little data on the effect of the ether group of the ester on activity as all were done using a fluorescene based system. It would be helpful to see some evidence that these selectivities extend to more real-world systems.

We thank the reviewers for this comment. We believe that this work is novel in that it isolates the activity of serum esterases on specific esters and enables one to design selective ester promoieties. We are actively synthesizing prodrugs of HEX (described in manuscript) and another series of phosphonate compounds (fosmidomycin and structural analogs) for additional study in more real-world systems.

4) It would be valuable to give an idea of how many opportunities there are for this and the spectrum of application. How many drug candidates are these for this? Permeability into Gram-positive organisms such as *S. aureus* is usually not much of an issue like it is in Gram-negatives. Outside of a few phosphonates, there aren't many that come to mind. This is also true of organisms, how broad-spectrum would one of these agents be? Perhaps the authors could show the activity of HEX against several pathogens.

We agree with the reviewers that development of novel antibacterials to combat Gram-negative bacteria is also a pressing concern. Creation of prodrugs that successfully mask phosphonate moieties in vivo expands the chemical inhibitor space that can be used to inhibit targets with highly charged active sites (such as many metabolic enzymes). We have previously found that prodrugging improves cellular potency against *P. falciparum* malaria parasites and *Mycobacterium tuberculosis*, both of particular public health concern (PMIDs: 28827774, 27676224). We believe that our studies outline a generalizable approach to addressing improving potency of highly polar compounds through specific prodrugging, which may be useful for those organisms as well. Examples of other anti-infective compounds in development where such prodrugging may be of use include the following references: PMIDs 31567411, 33361818, 31483651, 30730737, and reviewed in 31469328.

Our previously published studies give a strong indication as to the antimicrobial spectrum of prodrugs modified with lipophilic esters like POM, as we have previously characterized the activity of a series of analogs that target the 2-C-methyl-D-erythritol 4-phosphate pathway for isoprenoid biosynthesis. Similar to HEX, these “MEPicide” compounds possess a phosphonate warhead that requires active transit to pass the cell membrane. Non-prodrug (parent) analogs of these inhibitors readily engage and inhibit the highly conserved antimicrobial target enzyme (DXR) from multiple organisms in vitro and therefore have a broad antimicrobial spectrum. However, POM-prodrug variants of these MEPicides have markedly reduced antibacterial efficacy against many Gram-negative organisms (PMID: 32497104). Tested organisms included the following: Escherichia coli, Klebsiella pneumoniae, Salmonella typhimurium, Shigella sonnei, Serratia marcescens, and Burkholderia thailandensis. Notable exceptions are Francisella novicida and Mycobacterium tuberculosis (PMIDs: 23077474, 22024034). These previously published findings are echoed in our preliminary studies of POM-HEX (outside the scope of this current manuscript that focuses on S. aureus), in which we find that POM-HEX and POM-SF are much more potent against S. aureus than HEX or SF, but the reverse is true for *E. coli*. We believe that either POM-containing prodrugs are excluded from most Gram-negatives or they are not activated by their intracellular esterases, and in response to the reviewers’ concern, we now include these hypotheses in the discussion (lines 478 – 485).

Reviewer #2 (Recommendations for the authors):Antimicrobial resistance is a serious problem and S. aureus infections are also still a serious public health threat. The authors presented an interesting approach to the use of prodrugs against *S. aureus* infections. The research paper is very solidly based on scientific results. The research behind this article is very extensive and the basic research presented is very interesting and offers the possibility of developing new antibiotics. I will give more of a perspective as a researcher from antibacterial drug discovery. The advantage of the study is that both crystal structures are also available, allowing structure-based design of antibiotics that could be hydrolyzed by the proposed enzymes.– This work lacks a proof-of-concept to confirm the use of the proposed ester groups on real antibiotic molecules and to confirm hydrolysis in *S. aureus* by the two proposed enzymes FrmB and GloB.

We thank the reviewers for this comment. We believe that this work is novel in that it isolates the activity of serum esterases on specific esters and enables one to design selective esters. We are actively synthesizing prodrugs of HEX (described in manuscript) and another series of phosphonate compounds (fosmidomycin and structural analogs) for additional study in more real-world systems.

– It is not clear for which functional groups this approach would be most appropriate, hydroxyl, carboxyl?

To date we have primarily investigated phosphate and phosphonate antibiotics where attachment of a promoiety to the phosph(on)ate oxygen(s) is beneficial. We hypothesize that this is due to occlusion of the charge, which facilitates transit across the Gram-positive membrane (PMID: 32497104) Ultimately, the limitations to this approach may be determined by the route of chemical synthesis used for the inhibitors in question.

– With the introduction of ester groups, the logP is increased and so are inappropriate physicochemical properties. Due to the increased logP, plasma protein binding could be high, thus reducing in vivo efficacy.

We agree with the reviewer that lipophilic ester prodrugging strategies require attention to logP, although many successful anti-infectives and anti-parasitics do not follow’s Lipinski’s Rules as closely as other therapeutics (see commentary by McKerrow and Lipinski, PMID: 28623818). However, small molecule substrate-mimetic phosphonates (e.g., fosmidomycin, PubChem CID 572; and SF2312, PubChem CID 52913330) have excellent engagement of their protein targets in vitro but poor drug-like properties (logP < -2); in such cases, lipophilic ester prodrugging may improve both bioavailability, cellular permeability, and PK, thus improving in vivo efficacy. In fact, POM prodrugging has been successfully used for a number of FDA-approved antimicrobials, including cefditoren pivoxyl, sulbactam pivoxil, and adefovir dipivoxil.

– The last comment refers to the proposed approach for Gram-positive bacteria, where crossing antibiotics through the bacterial cell wall is usually not so problematic. The approach would be much more interesting and necessary for Gram-negative bacteria.

We agree with the reviewers that development of novel antibacterials to combat Gram-negative bacteria is also a pressing concern. Creation of prodrugs that successfully mask phosphonate moieties in vivo expands the chemical inhibitor space that can be used to inhibit targets with highly charged active sites (such as many metabolic enzymes). We have previously found that prodrugging improves cellular potency against *P. falciparum* malaria parasites and *Mycobacterium tuberculosis*, both of particular public health concern (PMIDs: 28827774, 27676224). We believe that our studies outline a generalizable approach to addressing improving potency of highly polar compounds through specific prodrugging, which may be useful for those organisms as well. Examples of other anti-infective compounds in development where such prodrugging may be of use include the following references: PMIDs 31567411, 33361818, 31483651, 30730737, and reviewed in 31469328.

Our previously published studies give a strong indication as to the antimicrobial spectrum of prodrugs modified with lipophilic esters like POM, as we have previously characterized the activity of a series of analogs that target the 2-C-methyl-D-erythritol 4-phosphate pathway for isoprenoid biosynthesis. Similar to HEX, these “MEPicide” compounds possess a phosphonate warhead that requires active transit to pass the cell membrane. Non-prodrug (parent) analogs of these inhibitors readily engage and inhibit the highly conserved antimicrobial target enzyme (DXR) from multiple organisms in vitro and therefore have a broad antimicrobial spectrum. POM-prodrug variants of these MEPicides have markedly reduced antibacterial efficacy against many Gram-negative organisms (PMID: 32497104), including Escherichia coli, Klebsiella pneumoniae, Salmonella typhimurium, Shigella sonnei, Serratia marcescens, and Burkholderia thailandensis. However, POM-prodrugging is effective in improving cellular potency against Francisella novicida and Mycobacterium tuberculosis (PMIDs: 23077474, 22024034). These previously published findings are echoed in our preliminary studies of POM-HEX (outside the scope of this current manuscript that focuses on S. aureus), in which we find that POM-HEX and POM-SF are much more potent against S. aureus than HEX or SF, but the reverse is true for *E. coli*. We believe that either POM-containing prodrugs are excluded from most Gramnegatives or they are not activated by their intracellular esterases, and in response to the reviewers’ concern, we now include these hypotheses in the discussion (lines 478 – 485).